# PyESPERv1.0.0: A Python implementation of empirical seawater property estimation routines (ESPERs)

Larissa M. Dias<sup>1,2</sup>, Brendan R. Carter<sup>2</sup>

<sup>1</sup>Cooperative Institute for Climate, Ocean, and Ecosystem Studies, University of Washington, Seattle, 98105, USA <sup>2</sup>NOAA Pacific Marine Environmental Laboratory, Seattle, 98115, USA

Correspondence to: Larissa M. Dias (lmdias@uw.edu)

**Abstract.** This project produced a Python language implementation of locally interpolated regression (LIR) and neural network 10 (NN) algorithms from empirical seawater property estimation routines (PvESPERv1.0.0). These routines estimate total alkalinity, dissolved inorganic carbon, total pH, nitrate, phosphate, silicate, and oxygen from geographic coordinates, depth, salinity, and 16 combinations of 0 to 4 additional predictors (temperature and biogeochemical information), and were previously available only in the MATLAB programming language. Here we document modifications to reduce discrepancies between the implementations, calculate the disagreements between the methods, and quantify Global Ocean Data Analysis 15 Project (GLODAPv2.2022) reconstruction errors with PvESPER. While the PvESPER routine based on neural networks (PyESPER NN) faithfully reproduces the corresponding MATLAB routine estimates of properties that do not require anthropogenic carbon change information, PyESPER LIR and—to a lesser extent—PyESPER NN estimates for total pH and dissolved inorganic carbon do not exactly reproduce the MATLAB routine estimates due to differences in interpolation and extrapolation methods between the programming languages. While the MATLAB and Python LIR-based estimates are not 20 identical, we show that they are similarly skilled at reproducing the GLODAPv2.2022 data product and are thus comparable. This project increases the accessibility of ESPERv1.01.01 algorithms by providing users with code in the freely available Python language and enables future ESPER updates to be released in multiple coding languages.

#### 1 Introduction

- Ship-based biogeochemical data, as compiled within the Global Ocean Data Analysis Project (GLODAP; Lauvset et al., 2022)

  have high precision and accuracy, but are seasonally biased and spatially sparse (Hauck et al., 2023). International efforts to deploy biogeochemical (BGC) profiling floats with broad spatial coverage and high temporal resolution (10 days) are ongoing (Bittig et al., 2019), with potential to greatly augment available ocean carbon cycle and biogeochemical data. These data can then support a wide variety of research topics and management applications (e.g., warming, acidification, eutrophication, deoxygenation, fisheries, and ecosystem studies). This strategy leverages the high precision and accuracy of ship-based measurements to calibrate and validate the BGC float sensors periodically throughout a float deployment. To do this, machine learning and regression algorithms—which take advantage of the strong regional correlations between seawater properties in the open ocean, and especially the ocean interior (Bittig et al., 2018; Carter et al., 2017, 2021)—are used to map the ship-based information onto "reference depth" portions of the float profiles.
- The empirical seawater property estimation routines (ESPERv1.01.01, henceforth referred to as ESPERs), originally written in MATLAB programming language, aim to help realize the full potential of BGC float data by using machine learning techniques and regression strategies to predict total alkalinity (TA), dissolved inorganic carbon (DIC), pH on the total scale (pH<sub>T</sub>), phosphate, nitrate, silicate, and oxygen from commonly measured physical and BGC parameters (Carter et al., 2021).

The algorithms are used to calibrate float profiles (Maurer et al., 2021). In addition, since two carbonate system property

40 measurements are necessary to fully quantify the carbonate system in seawater (Zeebe and Wolf-Gladrow, 2001) and BGC floats only have the capability to measure pH<sub>T</sub>, these algorithms have the potential to provide (calculated) TA or DIC as a secondary constraint for the marine carbonate system. This method also offers an alternative to using models to estimate variables for carbonate chemistry calculations when nutrient information is unavailable, which potentially has high error values. ESPERs have also been used to map ship-based information across spatial and temporal scales for other applications including estimation of TA for adjustments of pH and fugacity of CO<sub>2</sub> (fCO<sub>2</sub>) to in situ conditions for data products (Jiang et al., 2021), and estimation of TA and seawater properties necessary for estimation of ocean acidification indicators (Jiang et al., 2020; Sharp et al., 2024). Recent research has also shown that similar machine learning estimation algorithms have potential for the development of four-dimensional data products such as the Gridded Ocean Biogeochemistry from Artificial Intelligence – Oxygen (GOBAI-O<sub>2</sub>; Sharp et al., 2023) and the Mapped Observation-Based Oceanic DIC (MOBO-DIC; Keppler et al., 2020).

## 1.1 Importance

Tanhua et al. (2021) and others have argued that researchers should utilize workflows that produce findable, accessible, interoperable, and reusable (FAIR) data products. ESPERs are publicly available (findable) on Zenodo, with updates published to GitHub, free (accessible), and provide the option for users to cite a digital object identifier (DOI) for each version (reusable). However, until now ESPERs were only available in the proprietary MATLAB programming language, which posed a barrier to accessibility and interoperability that we aim to address. Future updates may include even more accessible features such as a user interface.

## 1.2 Goals

This project aimed to create a freely available Python implementation of ESPERs (PyESPERv1.0.0, henceforth referred to as PyESPERs; Carter et al., 2021; Dias and Carter, 2025) that is equivalent to the MATLAB version within ±2 × Estimate Uncertainties (σ) for all estimated biogeochemical properties (TA, DIC, pH<sub>T</sub>, nitrate, phosphate, silicate, and oxygen). PyESPER code is freely available at Zenodo and updates will be made available at the GitHub repository (*see* Sect. "Code availability").

## 2 Methods

ESPER algorithms were translated into Python coding language, while associated files were either translated into Python or read by Python as MATLAB files. Some original methods were required to allow interpolations to be similar in Python to those of MATLAB ESPERs.

## 2.1 ESPERs

ESPERs allow estimation of biogeochemical seawater properties using coordinates, depth, salinity, and other optional inputs from a single function call. While sharing a similar set of equations and required input data, ESPERs have two variants that use locally interpolated regressions (ESPER\_LIR) and neural networks (ESPER\_NN), respectively, along with a mixed estimate (ESPER\_Mixed) that is the mean of estimates from the two functions (Carter et al., 2017). There are a couple of reasons to maintain the separate ESPER LIR, NN, or Mixed options, from an end-user perspective, and these reasons are also true for PyESPERs.

- 1. ESPER\_LIRs predate the ESPER\_NNs and have been used as a standalone data product for various research purposes (see Carter et al., 2016; Carter et al., 2018). Long-term users of these LIRs have previously expressed desire for consistency between versions (e.g., when depth was taken out as predictor for pH<sub>T</sub>), and some of them already use CANYON-B (Bittig et al., 2019) as a neural net option for comparison. Therefore, these users who desire consistency would most likely prefer to use ESPER LIR.
- ESPER\_LIRs are more transparent than ESPER\_NN, as it is simple to parse apart coefficients at the gridded locations and to see how the equations are a result of these. ESPER\_LIRs also rely on a grid, which may appeal to some users.
  - 3. ESPER\_NNs work a bit better on average than ESPER\_LIRs and work more like a mapping product in that 3D coordinates are predictors, which may alternately appeal to some users.
- 4. Although the ESPER\_Mixed estimates perform better on average than LIRs or NNs do independently, there are cases where they have greater bias and RMSE than LIRs or NNs (e.g., when using equations 1-3 for phosphate or nitrate at all depths; Carter et al., 2021). Users may want to assess each scenario independently and choose which method is most appropriate according to their needs.
  - 5. The NNs are more closely reproduced between the MATLAB and Python ESPER implementations.

## 90 2.1.1 Locally interpolated regressions

The most recent versions of ESPER\_LIRs (version 1.01.01; version 3 of LIRs) use a standard set of equations of the format shown by Eq. (1) to estimate up to seven different biogeochemical water properties using up to 16 equations with different combinations of input parameters (*see* Sect. "*Appendix A*", Tables A1 and A2; Carter et al., 2021):

$$X = C_0 + \sum_{i=1}^{n} C_i P_i \tag{1}$$

where X is the estimated property (TA, DIC, pH<sub>T</sub>, nitrate, phosphate, silicate, or oxygen),  $C_0$  is the intercept, and  $C_i$  is the coefficient for each of the n predictors  $P_i$ . The intercepts ( $C_0$ ) and coefficients ( $C_i$ ) vary with location (latitude, longitude, and depth) and are different for each of the predictor variables ( $P_i$ ; Tables A1 and A2; Carter et al., 2021). The most recent ESPERs were trained and assessed on the GLODAPv2.2020 (Olsen et al., 2020) data product, which includes data from 946 cruises

and spanning 1972–2019, and additional data sets from the Mediterranean Sea and Gulf of Mexico (Carter et al., 2021, Supplementary Information) taken from the Coastal Ocean Data Analysis Project (CODAP, Jiang et al., 2021) and the CARIMED data product (Álvarez et al., 2019).

When the ESPER\_LIR function is called, the routines interpolate a pre-determined grid of C's (intercepts and coefficients) to user-defined locations. Linear interpolation is used within the grid and for extrapolation, and this method utilizes an underlying Delaunay triangulation with MATLAB's scatteredInterpolant function (Carter et al., 2021). The three-dimensional interpolation algorithm is implemented differently in MATLAB and Python, and although both calculations are valid, this difference in implementation is the source of disagreements we find and later quantify between ESPER and PyESPER.

105

ESPER LIR coefficients have been determined on a grid using a moving window regression strategy similar to the approach 110 first outlined by Velo et al. (2013), resulting in a set of intercept and coefficient estimates for each of 16 equations for 7 possible properties at 44,957 total locations on a 5° latitude (-84.5°-85.5° N) x 5° longitude (-19.5°-375.5° E) x 33 depth (0-5500 m) ocean interior grid subsampled from the World Ocean Atlas gridded product (Carter et al., 2016, 2017, 2021). These coefficients were fit using regressions relating the property of interest (X) to different combinations of up to five predictor properties (P, Tables A1 and A2), relating to each possible equation as in Eq. (1). Depth (scaled to  $\frac{1}{2}$ ) is included 115 as a coordinate for coefficient interpolation, but depth is not used as a predictor for the current ESPER version (it was included in an earlier version, but only when predicting pH<sub>T</sub>; Carter et al., 2017). Data for each regression fit are selected from "windows" of data that are within 15° latitude, 30°/cosine(latitude) in longitude, and within either (100 + z/10) m depth or 0.1 kg m<sup>-3</sup> of the estimated density of seawater at that coordinate location, where z is depth in m (Carter et al., 2021). If either the depth-based or the density-based criterion applies, data are selected for that location, which allows water masses to 120 impact window selection along with depth. If fewer than 100 measurements fall within a window, the dimensions are doubled. In LIRv2, windows were iteratively scaled by a factor of the iteration number until at least 100 measurements are selected to train each regression (Carter et al., 2017). For ESPER LIRs (LIRv3), it is argued that increasing window size has the following benefits: (1) includes more data for regression fits, (2) introduces more modes of oceanographic variability into fitting data, and (3) reduces multicollinearity (Carter et al., 2021). However, the risk of increasing window size is that they 125 will be less appropriate locally. A weighting term is applied to help account for this by reducing to cost of regression misfits to data that are distant or at significantly different depths from the location, with a cap to prevent overfitting to nearby coordinates (see Carter et al., 2021). Regression coefficients ( $C_0$  and  $C_i$ ) are then fit using Eq. (2), with separate regressions for northern hemisphere Atlantic, Mediterranean, and Arctic, and other global locations, to prevent interpolation across Central America or the Bering Strait.

$$XW = \left(C_0 + \sum_{i=1}^n C_i P_i\right) W \tag{2}$$

PyESPER\_LIR does not duplicate this portion of the effort but instead builds directly upon the grid of coefficients obtained for and utilized by the MATLAB implementation of ESPER\_LIR.

- 135 When the function is called, ESPER LIR uses MATLAB's scatteredInterpolant (linear interpolation and extrapolations) function to interpolate this previously-created grid of regression coefficients to the user-provided set of coordinates, resulting in coefficient estimates at the desired locations (Carter et al., 2021). This method uses a Delaunay triangulation of the scattered sample points to perform interpolations and extrapolations. Different valid mathematics can be used to obtain these Delaunay triangulations and to extrapolate and interpolate, and efforts to identify a Python method for these tasks that exactly replicated 140 MATLAB results were unsuccessful. The most similar and least computationally intensive results to those of MATLAB's scatteredInterpolant were produced by combining Python's scipy package functions LinearNDInterpolator (interpolate subpackage) and Delaunay (spatial subpackage; Virtanen et al., 2020). However, since LinearNDInterpolator does not extrapolate, and other Python functions did not produce similar results to those of MATLAB when using similar methods (see Appendix D), the gridded set of three-dimensional coordinates (44,957 locations based on the World Ocean Atlas) and 145 corresponding coefficient estimates provided by ESPER LIRs were expanded in MATLAB to 106,400 locations on a grid with estimates every 5° latitude (-94.5°-90.5° N) and longitude (-19.5°-375.5° E) and up to 9000 m depth and applied to scatteredInterpolant within ESPER LIR to provide coefficient estimates for the external locations through extrapolation. This grid, with equivalent coefficients within the original parts of the grid and extrapolations outside of the grid, was read in Python when LIRs were called. The expanded grid allowed Python functions to avoid extrapolations and rely solely on interpolation 150 and triangulation methods when estimating coefficients at user-defined locations. While some of these locations are unphysical (e.g.,  $\pm > 90^{\circ}$  N or on land), the coefficients nevertheless provide valid extrapolations from MATLAB for the full possible domain that can then be interpolated in PyESPER LIR. PyESPER LIR otherwise replicated ESPER LIR's separation of data from the Atlantic Ocean, Mediterranean Sea, and Arctic Ocean and data from the Indo-Pacific and Southern Ocean regions.
- During the creation of this expanded grid, a grouping error was observed in current versions of MATLAB ESPER\_LIRs. Specifically, the mirrored portion of the grid found at < 0° E and > 360° E and north of 40° S are not correctly flagged as belonging to the Atlantic grid. The practical effect of this bug was that estimates near the Prime Meridian and near the cutoff between the Southern Ocean and the Atlantic Ocean had extrapolated coefficients instead of interpolated coefficients. This bug was fixed for both MATLAB ESPER\_LIR and PyESPER\_LIR comparisons for this paper, and a fixed grouping routine is now provided at the original MATLAB ESPER repository with corresponding documentation and will be included in future updates to ESPER\_LIRs.

#### 2.1.2 Neural networks

ESPER\_NNs use feed-forward neural networks with latitude, depth, cosine(longitude-20°E), cosine(longitude-110°E) and the parameters from Table A2 as predictors. Four neural networks are used in each of the two ocean regions, which are the same as those used for LIRs (Atlantic-Mediterranean-Arctic and Indo-Pacific-Southern), resulting in 896 total neural networks (8 for each of 16 combinations of predictors for 7 property estimates; Carter et al., 2021). An ensemble of four previously-created neural networks with different combinations of neurons and hidden layers, including a single one-hidden-layer network with 40 neurons and three two-hidden-layer networks with 30/10, 25/15, and 20/20 neurons in the first/second hidden layers is used to minimize the impact of errors from any one neural network (Carter et al., 2021).

In ESPER\_NN the neural networks are encoded as functions to avoid requiring access to the Machine Learning toolbox within MATLAB. Here we further translate these functions to Python. The resultant Python functions replicate the functions in ESPER\_NN to within machine precision. ESPER\_NNs linearly interpolate between the two regions of neural networks by latitude across the Southern Atlantic Ocean and Bering Sea and between the North Pacific and Arctic Oceans. Zonal transitions in the Southern Atlantic and Indo-Pacific-Southern Ocean network are also implemented. This interpolation uses custom-written 1 or 2D interpolations that are handled identically in both programming environments.

### 2.1.3 Mixed estimates

The mixed estimate for each input location is the mean of the LIR and NN estimates and therefore is trivially reproduced by a simple single function call within Python.

## 180 2.1.4 Anthropogenic carbon

The impacts of anthropogenic carbon ( $C_{ant}$ ) are approximated in ESPER and PyESPER using a 1° x 1° gridded transit time distribution (Waugh et al., 2006)-based  $C_{ant}$  product referenced to the year 2002 (Lauvset et al., 2016). ESPERs assume that oceanic  $C_{ant}$  increases proportionally to atmospheric anthropogenic  $CO_2$  (transient steady state assumptions; Gammon et al., 1982; Gruber et al., 2019; Tanhua et al., 2007). This implies that the "shape" of the  $C_{ant}$  vertical profile (gradient) remains constant with continuous exponential increases of atmospheric  $CO_2$  and ocean  $C_{ant}$  according to Eq. (3; Carter et al., 2021).

$$C_{ant\_year\_location} = C_{ant\_year\_location} e^{0.018989(year-2002)}$$
(3)

The coefficient in Eq. (3) is derived from Gruber et al.'s (2019) assumption of a 28% increase in  $C_{\rm ant}$  from 1994–2007, and enables estimating  $C_{\rm ant}$  for a location in a desired year when  $C_{\rm ant}$  is known for that same location in a reference year (2002;

Carter et al., 2021). This approach does not allow for non-steady-state variations, which is accounted for in overall uncertainty estimates, and is noted as a significant source of uncertainty for projections beyond ~2030.

ESPERs were trained on data for pH<sub>T</sub> and DIC which were transformed to the year 2002, then modified back to the original measurement dates using Eq. (3). ESPERs and PyESPERs estimate the  $C_{ant}$  component of DIC and pH<sub>T</sub> in output variables for 2002 by interpolating the 2002  $C_{ant}$  grid to user-provided coordinates and then applying Eq. (3) to estimate  $C_{ant}$  for the user-requested estimate year. As with original ESPERs, this method is not meant to be used when  $C_{ant}$  is of primary interest, but rather provides a means of quickly adjusting DIC or pH<sub>T</sub> to a reference year (Carter et al., 2021). Likewise, these methods are not adequate for making reliable projections beyond the year 2030, or perhaps sooner in coastal or other areas where the underlying global open-ocean anthropogenic carbon estimations have greater uncertainties (Carter et al., 2021).

#### 2.2 Uncertainty estimation

ESPERs and PyESPERs return depth- and salinity-dependent uncertainties for each property at the  $1\sigma$  (one standard uncertainty) level, meaning approximately 95% of new open-ocean measurements from GLODAPv2.2022 should fall within  $\pm$  twice the ESPER uncertainties (Carter et al., 2021). As in Carter et al. (2021), baseline error estimates in depth and salinity space ( $E_{X\_Est}$ ) are interpolated based on root mean square errors (RMSEs) of all predictions from validation versions of the routines within bins of salinity and depth. ESPER\_LIRs and PyESPER\_LIRs scale these uncertainties using user-provided predictor uncertainty estimates ( $E_{Pi\_Provided}$ ). Eq. (4) is used when user-provided uncertainties exceed default assumed input uncertainties ( $E_{Pi\_Default}$ ; Table A3):

$$E_{X\_Output} = \sqrt{E_{X\_Est}^2 - \sum_{i=1}^n \left(\frac{\partial X}{\partial P_i} E_{Pi\_Default}\right)^2 + \sum_{i=1}^n \left(\frac{\partial X}{\partial P_i} E_{Pi\_Provided}\right)^2}$$
(4)

where  $\frac{\partial X}{\partial P_i}$  is the sensitivity of the property estimate X to the  $i^{th}$  predictor  $P_i$ . ESPER\_NNs and PyESPER\_NNs estimate sensitivities by iteratively perturbing the input predictors if the user specifies uncertainties that are larger than default. Mixed uncertainties are the minimum uncertainties assessed for LIR and NN estimates.

## 2.3 Assessment

210

215

For many applications, the most critical validation is a test of the reconstruction of withheld data. However, such an exercise requires training alternative versions of the method after withholding data, and, as of now, PyESPER is not separately trained, but is instead reliant on the ESPER training that was performed and validated previously with MATLAB (Carter et al. 2021). For this publication, we aim to instead show that PyESPER and ESPER provide quantitatively similar results and assert that the validation presented earlier for ESPER in MATLAB can be considered to also be appropriate for PyESPER in all but a

limited number of specific exceptional cases. To support this claim, PvESPER and ESPER were used to estimate values for the GLODAPv2.2022 data product (1,381,248 sets of measurements; Fig. 1) with each equation and output variable combination. This dataset included a wide range of input data, and comparison of PyESPER and ESPER was primarily 220 considered from application to the high-quality "open ocean" (a) portion of the GLODAP dataset as in Carter et al. (2021), defined as GLODAP data with only World Ocean Circulation Experiment (WOCE) data quality control flag categories of 2 (Acceptable) and secondary quality control flag categories of 1 (subjected to full secondary quality control) for all possible input and measurement data, and for salinities between 30-37 (n=306,227 for TA, 343,580 for DIC, 199,304 for pH<sub>T</sub>, and 764,301 for phosphate, nitrate, silicate, and oxygen). Additional comparison with the entire GLODAPv2.2022 dataset ("whole 225 ocean" or w), including NaNs and anomalous data with salinities <30 and temperatures <0 °C, which are not recommended for use with ESPERs, is presented in Sect. "Appendix B". These comparisons are used as a rigorous test of the fidelity of the PvESPER estimates to the ESPER estimates. Resulting estimates were compared graphically and with normalized root mean square error (RMSE<sub>n</sub>; equivalent to RMSE divided by the mean of the MATLAB estimate for each variable) for each equation case globally and regionally, and across depths. RMSE<sub>n</sub> was used because it allows for comparison between variables of 230 different scales. Additionally, where measured values were present in the dataset, both ESPER and PyESPER were validated against the measured data, though, again, this is not a validation of the method as much as a check that both variants provide similar values.

#### 2.3.1 DIC application

As an additional comparison of the LIR method differences, DIC estimates from both PyESPER\_LIR and ESPER\_LIR were applied to the Roemmich and Gilson Argo-derived climatology (Roemmich and Gilson, 2009) to create mapped annual surface estimates of DIC.

Figure 1: Location of GLODAPv2.2022 data used to compare PyESPER to MATLAB ESPER estimates (a), and histograms of the distributions of measured GLODAPv2.2022 variables used as inputs for PyESPERv1.0.0 and ESPER algorithms (b-g).

#### 3 Results and Discussion

PyESPER and ESPER produced open ocean estimates with mean differences (Python estimate – MATLAB estimate) of <±0.04 for all parameters, and NNs had smaller mean differences of <±0.004 for all parameters (units are μmol kg<sup>-1</sup> except for pH<sub>T</sub>) estimated from open ocean GLODAPv2.2022 data, although the standard deviations of these differences and uncertainties associated with estimates were at times larger than the mean differences (Tables 1 and 2). The greatest RMSE<sub>n</sub> was 2.08x10<sup>-2</sup> for silicate estimates using LIRs. PyESPER\_NN functioned as an equivalent data product to ESPER\_NN for all data. For open ocean data, PyESPER\_LIRs functioned similarly to ESPER\_LIRs, with a large majority of identical estimates produced between the two data products.

## 3.1 Data product validation

260

250 Results of comparisons between MATLAB ESPERs and PyESPERs are described below.

#### 3.1.1 Locally interpolated regressions

When compared to the ESPER\_LIR results for the open ocean (o) GLODAPv2.2022 dataset, all equation-case and desired outcome variable combinations from PyESPER (PyESPER\_LIR – ESPER\_LIR estimates) resulted in mean differences of <±0.04 (Table 1). Mean (±standard deviation; RMSE<sub>n</sub>) PyESPER – ESPER\_LIR differences for each property are shown in Table 1. The very wide range of input data resulted in a wide range of estimates from both ESPER\_LIRs and PyESPER\_LIRs for all variables (Table 1; Fig. 2; for w see Sect. "Appendix B", Fig. B1), representing the large range of biogeochemical property values that can be found in the oceans. PyESPER\_LIR and ESPER\_LIR results worked similarly well in predicting measured values at locations, even with the outlier and unusual input data used (see Table B1), suggesting that Python estimates, although not identical to MATLAB estimates for these interpolations, were equivalently valid reconstructions.

oxygen estimates (all units except pH<sub>T</sub> are µmol kg<sup>-1</sup>) for open ocean (a) data and all equations combined (n=13,384,096 for TA, 13,384,096 Table 1: Mean (standard deviation), maximum, minimum, and normalized RMSE (RMSE<sub>n</sub>), for differences between MATLAB and Python LIRs, ESPER\_LIR and measured values, and PyESPER\_LIR and measured values for TA, DIC, pHr, phosphate, nitrate, silicate, and for DIC, 13,384,096 for pH<sub>T</sub>, 13,384,096 for phosphate, 12,718,592 for nitrate, 12,640,896 for silicate, and 12,757,792 for oxygen).

| 6        |                         | I1)                  |                       | II                        |                         |                      | (((                      |                       | ( , (                   |                      |                       |                       |
|----------|-------------------------|----------------------|-----------------------|---------------------------|-------------------------|----------------------|--------------------------|-----------------------|-------------------------|----------------------|-----------------------|-----------------------|
|          |                         | Python – MATLAI      | MATLAB                |                           |                         | MATLAB.              | <b>MATLAB - Measured</b> |                       |                         | Python - Measured    | <b>Aeasured</b>       |                       |
|          | Mean                    | Max                  | Min                   | $RMSE_n$                  | Mean                    | Max                  | Min                      | RMSEn                 | Mean                    | Max                  | Min                   | RMSEn                 |
| ŧ        | -4.75x10 <sup>-4</sup>  | $6.44x10^{1}$        | $-7.03x10^{1}$        | 4.64x10 <sup>-4</sup>     | $2.71x10^{-1}$          | $8.13x10^{2}$        | $-1.69 \times 10^{2}$    | 2.72x10 <sup>-3</sup> | $2.70x10^{-1}$          | $8.13x10^{2}$        | $-1.73x10^{2}$        | 2.71x10 <sup>-3</sup> |
| IA       | (1.08)                  |                      |                       |                           | (6.34)                  |                      |                          |                       | (6.32)                  |                      |                       |                       |
|          | $3.39x10^{-2}$          | $2.01x10^{2}$        | $-2.61 \times 10^{2}$ | 7.29x10 <sup>-4</sup>     | -4.40x10 <sup>-1</sup>  | $6.20\mathrm{x}10^2$ | $-3.20 \times 10^{2}$    | $3.90 \times 10^{-3}$ | $-4.02x10^{-1}$         | $6.20 \times 10^{2}$ | $-3.16x10^{2}$        | $3.90x10^{-3}$        |
| DIC      | (1.60)                  |                      |                       |                           | (8.55)                  |                      |                          |                       | (8.47)                  |                      |                       |                       |
| I        | -5.65x10 <sup>-5</sup>  | $5.05x10^{-1}$       | $-3.77$ x $10^{-1}$   | 5.36x10 <sup>-4</sup>     | -2.51x10 <sup>-3</sup>  | $1.14x10^{0}$        | $-6.80 \times 10^{-1}$   | 2.86x10 <sup>-3</sup> | $-2.56x10^{-3}$         | $1.14x10^{0}$        | $-5.46x10^{-1}$       | 2.84x10 <sup>-3</sup> |
| рн       | $(4.24x10^{-3})$        |                      |                       |                           | $(2.24x10^{-2})$        |                      |                          |                       | $(2.23 \times 10^{-2})$ |                      |                       |                       |
| Phosp-   | $3.08 \times 10^{-4}$   | 1.65                 | -2.17                 | 8.44x10 <sup>-3</sup>     | -1.54x10 <sup>-4</sup>  | 2.90                 | $-3.12 \times 10^{0}$    | $3.90 \times 10^{-2}$ | $-1.61x10^{-4}$         | $2.57x10^{0}$        | $-3.50 \times 10^{0}$ | $3.61 \times 10^{-2}$ |
| hate     | $(1.41x10^{-2})$        |                      |                       |                           | $(6.21x10^{-2})$        |                      |                          |                       | $(6.09 \times 10^{-2})$ |                      |                       |                       |
|          | $2.20x10^{-3}$          | $1.89x10^{1}$        | $-4.13x10^{1}$        | $1.30 \mathrm{x} 10^{-2}$ | -5.43x10 <sup>-3</sup>  | $4.23x10^{1}$        | $-3.45 \times 10^{1}$    | $3.62 \times 10^{-2}$ | $-7.67$ x $10^{-3}$     | $3.04x10^{1}$        | $-4.24x10^{-1}$       | $3.43x10^{-2}$        |
| Nitrate  | $(3.08 \times 10^{-1})$ |                      |                       |                           | $(8.58 \times 10^{-1})$ |                      |                          |                       | $(8.14 \times 10^{-1})$ |                      |                       |                       |
| 21:00    | $2.27x10^{-2}$          | $5.92x10^{1}$        | $-5.85 \times 10^{1}$ | $2.08 \times 10^{-2}$     | -6.60x10 <sup>-2</sup>  | $8.42x10^{1}$        | $-2.08 \times 10^{2}$    | 5.57x10 <sup>-2</sup> | $-4.28x10^{-2}$         | $8.20x10^{1}$        | $-2.08x10^{2}$        | $5.25 \times 10^{-2}$ |
| Silicate | (1.18)                  |                      |                       |                           | (3.19)                  |                      |                          |                       | (3.01)                  |                      |                       |                       |
|          | $3.98x10^{-3}$          | $3.08\mathrm{x}10^2$ | $-2.31x10^{2}$        | $1.10 x 10^{-2}$          | $6.06 \times 10^{-2}$   | $3.28\mathrm{x}10^2$ | $-4.23 \times 10^{2}$    | $4.66 \times 10^{-2}$ | $6.32x10^{-2}$          | $3.21x10^{2}$        | $-3.75x10^{2}$        | $4.55x10^{-2}$        |
| Oxygen   | (2.11)                  |                      |                       |                           | (8.90)                  |                      |                          |                       | (8.70)                  |                      |                       |                       |

Figure 2: Difference between Python and MATLAB locally interpolated regression estimates (*y*-axis) compared to MATLAB estimates (*x*-axis) for open ocean (<sub>o</sub>) data and all equations combined for TA (a, 13,384,096 total estimates from all equations), DIC (b, 13,384,096 estimates), pH<sub>T</sub> (c, 13,384,096 estimates), phosphate (d, 13,384,096 estimates), nitrate (e, 12,718,592 estimates), silicate (f, 12,640,896), and oxygen (g, 12,757,792 estimates; *n*=306,227 for TA, 343,580 for DIC, 199,304 for pH<sub>T</sub>, and 764,301 for phosphate, nitrate, silicate, and oxygen). Units for all except pH<sub>T</sub> are in μmol kg<sup>-1</sup>. Top and bottom side histograms represent the distribution of the x and y axes, respectively. Note the differences in *x*- and *y*-axes scales. RMSE*n* is the normalized root mean square error, or the RMSE of all divided by the mean of all estimates.

PvESPER LIRs were within  $2\sigma$  (~95% of measurements should fall within this uncertainty level) for most ocean regions, with a few exceptions which occurred predominantly in coastal areas or deep waters near the edges of the original MATLAB grid (Figs. 3 and 4). Spatial patterns in distribution of outliers shown in Fig. 4 appear to reflect locations where more edge-of-grid biogeochemical measurements were collected (e.g., near coasts and in deep waters). Hence, these exceptionally different locations aligned well with places where coefficients were extrapolated in MATLAB for use in PvESPER LIRs, compared to interpolations with far away "dummy points" within MATLAB ESPER LIRs (see Sect. 2.1.1, "Locally interpolated regressions"; Figs. 3, 4, and 5; for w Fig. B2 and B3, and Appendix D). Within regions where MATLAB and Python were interpolating similarly, far outliers were uncommon (Figs. 3, 4, 5, B2, and B3). When ESPER LIR and PyESPER LIR were applied to temperature and salinity from the Roemmich and Gilson climatology for the year 2023 (Roemmich and Gilson, 2009), patterns of surface DIC distribution were similar with a few minor nuances (Fig. C1), Notably, low DIC estimates covered a broader spatial extent in the western equatorial Pacific and Indian Oceans for PyESPER LIR estimates, and PyESPER LIR appeared to have a slightly low bias in some places relative to ESPER LIR. Beyond these minor differences, the mapped DIC demonstrates the similarity of the data products' functionality in an applied setting. While ESPER LIR and PyESPER LIR do not produce quantitatively identical estimates, it should be noted that both routines perform similarly well at reconstructing the GLODAPv2.2022 data product (Table 1; for w Table B1). These routines should not be considered identical but are comparable.

#### 3.1.2 Neural networks

When compared to the ESPER\_NN results for the open ocean ( $_o$ ) GLODAPv2.2022 dataset, all equation-case and desired outcome variable combinations from PyESPER\_NN (PyESPER – ESPER\_NN estimates) resulted in mean differences of < $\pm$ 0.004 (Table 2), a much smaller difference than for LIR comparisons. Mean ( $\pm$ standard deviation; RMSE $_n$ ) offset for each property is shown in Table 2. Since a very wide range of input data were used, a wide range of estimates were produced from both ESPER\_NNs and PyESPER\_NNs for all variables (Fig. 6), representing the high variability that can be found in the oceans (especially coastal regions, some of which were included in the "open ocean" dataset due to having salinities between 30–37 and quality-controlled data). Both PyESPER\_NN and ESPER\_NN results were nearly identical, even when outlier results were obtained from unusual input data from environments where ESPERs are not recommended for use (for example, resulting in negative DIC estimates in Fig. B4; see also Table B2). The largest relative disagreements were found for DIC and pH<sub>T</sub>, though these disagreements remained small relative to measurement uncertainties. These minor offsets are attributed to the programming language differences in the interpolation of the  $C_{ant}$  adjustment, which is only applied to these two properties.

Figure 3: Map of differences between Python and MATLAB ESPER locally interpolated regression estimates (total estimates *n*=13,384,096 for TA (a), DIC (b), pH<sub>T</sub> (c), and phosphate, 12,718,592 for nitrate (d), 12,640,896 for silicate (e), and 12,757,792 for oxygen (f)) for the open ocean (o), where small blue circles represent differences <2 x uncertainties of the MATLAB estimates (*n*=13,344,924 for TA, 13,354,980 for DIC, 13,349,438 for pH<sub>T</sub>, 13,357,843 for phosphate, 12,688,861 for nitrate, 12,597,608 for silicate, and 12,721,483 for oxygen), and red circles represent differences >2 x uncertainties of the MATLAB estimates (*n*=39,172 for TA, 29,116 for DIC, 34,658 for pH<sub>T</sub>, 26,253 for phosphate, 29,731 for nitrate, 43,288 for silicate, and 36,309 for oxygen).

Figure 4: Map of locations and depths (colorbar) where differences between Python and MATLAB ESPER locally interpolated regression estimates are greater than 2 x the estimate uncertainties for the open ocean (o, n=13,344,924 for TA (a), 13,354,980 for DIC (b), 13,349,438 for pH<sub>T</sub> (c), 13,357,843 for phosphate (d), 12,688,861 for nitrate (e), 12,597,608 for silicate (f), and 12,721,483 for oxygen (g)).

Figure 5: Map of locations where MATLAB was interpolating (n=1,365,170, blue) and extrapolating (n=16,078, red) from the grid to GLODAPv2.2022 data (a) and depth of extrapolations (b).

## 3.1.3 Anthropogenic carbon estimates

Although inconsistencies in results occur between Python and MATLAB when interpolating (same issue noted in Sect. 2.1.4, "Anthropogenic carbon"), anthropogenic carbon ( $C_{ant}$ ) estimates were similar between the two versions of ESPER. This was demonstrated by differences in DIC and pH<sub>T</sub> estimates for NNs, which only interpolate when estimating the contribution of  $C_{ant}$  to estimates (Fig. 6). The next generation of ESPER updates will include a new method for estimating  $C_{ant}$  (Tracer-Based Rapid Anthropogenic Carbon Estimation, or TRACEv1; Carter et al., *submitted*), which uses neural networks and should eliminate the need for interpolation. Currently, when  $C_{ant}$  estimates are required, the results from PyESPER\_NNs remain functionally identical to those from ESPER\_NNs, despite minor offsets from the interpolation methods.

## 330 3.2 Speed of calculation

PyESPERs take considerably longer than ESPERs to produce estimates. On a MacBook Air using Python Jupyter Notebook with standard internet connection, PyESPER\_NN produced results 0–1500 x slower than ESPER\_NN, while PyESPER\_LIR produced results about 7–500 x slower than ESPER\_LIRs, with magnitude of the slowdown dependent upon the number of variable inputs and equation cases requested and number of estimates required (Table 3). ESPER\_NNs were the fastest to execute, and took <2 s for all time tests, even when large datasets and all variable-equation case scenarios were requested. ESPER\_LIRs were the next-fastest, requiring <33 s for all time tests, followed by PyESPER\_NNs, which typically required 5–15 s to execute, but required >1400 s (23 min) for running large datasets and all variable-equation case scenarios. PyESPER\_LIRs were the slowest, and typically required 22–500 s to execute, but the longest scenario required 7530 s (125 min; Table 3). It is possible that this code can be further optimized for speed in future updates.

335

Table 2: Mean (standard deviation), maximum, minimum, and normalized RMSE (RMSE<sub>n</sub>) are shown for three scenarios: (1) between Python - MATLAB NNs, (2) MATLAB ESPER\_NN - measured values, and (3) PyESPER\_NN - measured values. Separate rows exist for TA, DIC, pHr, phosphate, nitrate, silicate, and oxygen estimates. All units except pHr are µmol kg-1, and data are for open oceans (a) and all equations combined.

| •              |                         |                       |                        |                        |                         |                          |                 |                       |                         |                      |                       |                       |
|----------------|-------------------------|-----------------------|------------------------|------------------------|-------------------------|--------------------------|-----------------|-----------------------|-------------------------|----------------------|-----------------------|-----------------------|
|                |                         | Python - MATLAB       | MATLAB                 |                        | I                       | <b>MATLAB - Measured</b> | . Measured      |                       |                         | Python - Measured    | Measured              |                       |
|                | Mean                    | Max                   | Min                    | RMSEn                  | Mean                    | Max                      | Min             | RMSEn                 | Mean                    | Max                  | Min                   | RMSEn                 |
| ŧ              | -4.49x10 <sup>-12</sup> | 4.00x10-6             | -2.00x10 <sup>-6</sup> | 2.53x10 <sup>-12</sup> | $3.40 \times 10^{-1}$   | $8.15 \times 10^{2}$     | $-1.78x10^{2}$  | 2.24x10 <sup>-3</sup> | $3.40 \times 10^{-1}$   | $8.15x10^{2}$        | $-1.78 \times 10^{2}$ | 2.24x10 <sup>-3</sup> |
| IA             | $(5.89 \times 10^{-9})$ |                       |                        |                        | (5.21)                  |                          |                 |                       | (5.21)                  |                      |                       |                       |
| OIG.           | -3.01 x10 <sup>-3</sup> | 2.31                  | -3.69                  | 4.22x10 <sup>-5</sup>  | $-2.94 \times 10^{-1}$  | $6.17x10^{2}$            | $-3.37x10^{2}$  | $3.49 \times 10^{-3}$ | $-2.97$ x $10^{-1}$     | $6.18\mathrm{x}10^2$ | $-3.37$ x $10^{2}$    | 3.49x10 <sup>-3</sup> |
| DIC            | $(9.29 \times 10^{-2})$ |                       |                        |                        | (7.67)                  |                          |                 |                       | (7.67)                  |                      |                       |                       |
| ;              | $1.07 \times 10^{-5}$   | $5.60 \times 10^{-3}$ | $-7.65 \times 10^{-3}$ | $2.98 \times 10^{-5}$  | $-4.59$ x $10^{-3}$     | $4.71x10^{-1}$           | $-6.58x10^{-1}$ | $2.10x10^{-3}$        | $-4.58 \times 10^{-3}$  | $4.71x10^{-1}$       | -6.58x10              | 2.10x10 <sup>-3</sup> |
| $ m pH_T$      | $(2.34 \times 10^{-4})$ |                       |                        |                        | $(1.59 \times 10^{-2})$ |                          |                 |                       | $(1.59 \times 10^{-2})$ |                      | -                     |                       |
| Ē              | $-6.19x10^{-14}$        | $2.50 \times 10^{-8}$ | -1.25x10 <sup>-7</sup> | $1.31x10^{-10}$        | $1.15 \times 10^{-3}$   | 2.12                     | -2.81           | $3.06 \times 10^{-2}$ | $1.15x10^{-3}$          | 2.12                 | -2.81                 | 3.06x10 <sup>-2</sup> |
| rnosp-<br>hate | $(6.60 \times 10^{-})$  |                       |                        |                        | $(5.19 \times 10^{-2})$ |                          |                 |                       | $(5.19 \times 10^{-2})$ |                      |                       |                       |
| IIato          | (11)                    |                       |                        |                        |                         |                          |                 |                       |                         |                      |                       |                       |
|                | $-7.80 \times 10^{-13}$ | $1.35x10^{-7}$        | $-2.28x10^{-6}$        | $3.76x10^{-11}$        | $-2.24x10^{-3}$         | $4.06 \times 10^{1}$     | $-3.40x10^{1}$  | $2.93 \times 10^{-2}$ | $-2.24x10^{-3}$         | $4.06 \times 10^{1}$ | $-3.40 \times 10^{1}$ | $2.93 \times 10^{-2}$ |
| Nitrate        | (8.91x10 <sup>-</sup>   |                       |                        |                        | $(7.17 \times 10^{-1})$ |                          |                 |                       | $(7.17x10^{-1})$        |                      |                       |                       |
|                | 10)                     |                       |                        |                        |                         |                          |                 |                       |                         |                      |                       |                       |
| 0.11.00        | -1.24x10 <sup>-12</sup> | $2.11x10^{-7}$        | -2.97x10 <sup>-6</sup> | $3.42x10^{-11}$        | $4.96 \times 10^{-3}$   | $1.23 \times 10^{2}$     | $-8.29x10^{1}$  | $4.42 \times 10^{-2}$ | 4.96x10 <sup>-3</sup>   | $1.23 \times 10^{2}$ | $-8.29x10^{1}$        | 4.42x10 <sup>-2</sup> |
| Silicate       | $(1.98 \times 10^{-9})$ |                       |                        |                        | (2.55)                  |                          |                 |                       | (2.55)                  |                      |                       |                       |
|                | $-4.42x10^{-13}$        | $1.00 x 10^{-8}$      | $-1.00x10^{-7}$        | $1.06x10^{-12}$        | $5.33 \times 10^{-2}$   | $3.54 \times 10^{2}$     | $-2.06x10^{2}$  | $3.82 \times 10^{-2}$ | $5.33 \times 10^{-2}$   | $3.54x10^{2}$        | $-2.06 \times 10^{2}$ | $3.82 \times 10^{-2}$ |
| Oxygen         | (2.09x10                |                       |                        |                        | (7.29)                  |                          |                 |                       | (7.29)                  |                      |                       |                       |
|                | 10)                     |                       |                        |                        |                         |                          |                 |                       |                         |                      |                       |                       |

MATLAB estimates from NN

Figure 6: Difference between Python and MATLAB neural network estimates (y-axis) compared to MATLAB estimates (x-axis) for open ocean (o) data and all equations combined for TA (a, 4,899,512 total estimates from all equations), DIC (b, 5,497,004 estimates), pH<sub>T</sub> (c, 3,188,864 estimates), phosphate (d, 12,228,432 estimates), nitrate (e, 12,228,432 estimates), silicate (f, 12,228,432 estimates), and oxygen (g, 12,228,560 estimates; n=306,227 for TA, 343,580 for DIC, 199,304 for pH<sub>T</sub>, and 764,301 for phosphate, nitrate, silicate, and oxygen). Units for all except pH<sub>T</sub> are in µmol kg<sup>-1</sup>. Top and bottom side histograms represent the distribution of the x and y axes, respectively. Note the differences in x-and y-axes scales. RMSE<sub>n</sub> is the normalized root mean square error, or the RMSE divided by the mean of all estimates from MATLAB NN.

Table 3: Time required to produce estimates for PyESPERv1.0.0s and ESPERs (LIRs and NNs) for different desired variable, equation-case, and number of estimates scenarios.

|               |                    | Number of | PyESPER_NN | ESPER_NN | PyESPER_LIR | ESPER_LIR |
|---------------|--------------------|-----------|------------|----------|-------------|-----------|
| Variable      | <b>Equation(s)</b> | Estimates | time (s)   | time (s) | time (s)    | time (s)  |
| TA            | 1                  | 10        | 6.55       | 0.01     | 22.35       | 0.77      |
| TA            | 1                  | 100       | 5.87       | 0.01     | 19.98       | 0.60      |
| TA            | 2                  | 100       | 5.82       | 0.01     | 25.90       | 0.79      |
| TA            | 3                  | 100       | 5.79       | 0.01     | 22.82       | 0.81      |
| TA            | 4                  | 100       | 5.90       | 0.01     | 24.01       | 0.78      |
| TA            | 5                  | 100       | 5.80       | 0.00     | 23.60       | 0.75      |
| TA            | 6                  | 100       | 5.88       | 0.01     | 22.42       | 0.79      |
| TA            | 7                  | 100       | 5.88       | 0.00     | 23.03       | 0.78      |
| TA            | 8                  | 100       | 5.84       | 0.00     | 22.51       | 0.80      |
| TA            | 9                  | 100       | 5.87       | 0.00     | 22.42       | 0.81      |
| TA            | 10                 | 100       | 5.82       | 0.01     | 22.60       | 0.74      |
| TA            | 11                 | 100       | 5.84       | 0.00     | 22.28       | 0.74      |
| TA            | 12                 | 100       | 5.90       | 0.00     | 22.43       | 0.75      |
| TA            | 13                 | 100       | 5.88       | 0.00     | 22.37       | 0.79      |
| TA            | 14                 | 100       | 5.82       | 0.01     | 22.46       | 0.77      |
| TA            | 15                 | 100       | 5.81       | 0.00     | 22.35       | 0.84      |
| TA            | 16                 | 100       | 5.81       | 0.01     | 22.57       | 0.74      |
| TA            | 1-16               | 100       | 11.06      | 0.04     | 312.13      | 0.62      |
| TA            | 1                  | 1000      | 11.50      | 0.03     | 29.69       | 0.76      |
| TA            | 1                  | 10,000    | 61.54      | 0.12     | 57.59       | 0.83      |
| TA            | 1                  | 100,000   | 950.78     | 0.62     | 325.87      | 1.55      |
| DIC           | 1                  | 100       | 5.86       | 1.55     | 32.51       | 2.69      |
| DIC           | 1-16               | 100       | 10.86      | 1.53     | 365.58      | 1.54      |
| рН            | 1                  | 100       | 6.09       | 0.06     | 54.65       | 0.81      |
| рH            | 1-16               | 100       | 15.37      | 0.46     | 766.74      | 3.41      |
| Phosphate     | 1                  | 100       | 5.85       | 0.01     | 23.46       | 3.39      |
| Phosphate     | 1-16               | 100       | 11.01      | 0.06     | 376.30      | 0.80      |
| Nitrate       | 1                  | 100       | 5.85       | 0.01     | 23.07       | 0.74      |
| Nitrate       | 1-16               | 100       | 11.04      | 0.05     | 364.13      | 3.56      |
| Silicate      | 1                  | 100       | 5.84       | 0.02     | 26.84       | 3.64      |
| Silicate      | 1-16               | 100       | 11.02      | 0.04     | 365.34      | 0.82      |
| Oxygen        | 1                  | 100       | 6.97       | 0.01     | 24.60       | 0.78      |
| Oxygen        | 1-16               | 100       | 10.98      | 0.04     | 385.28      | 2.15      |
| All Variables | 1                  | 100       | 11.81      | 0.01     | 194.31      | 13.86     |
| All Variables | 1                  | 10,000    | 147.26     | 0.10     | 561.29      | 15.17     |
| All Variables | 1-16               | 100       | 49.53      | 0.09     | 3182.56     | 15.26     |
| All Variables | 1-16               | 10,000    | 1443.63    | 1.67     | 7530.23     | 32.13     |

#### 3.3 Future improvements

360 Updated ESPERs will be trained and assessed using GLODAPv2.2023 (or later versions), which includes 1108 cruises (compared to 946 cruises from GLODAPv2.2020, the current data product used. Additionally, future ESPERs will incorporate depth (z) as an optional predictor variable for consistency with LIPHR, a prior version for estimating pH<sub>T</sub> (Carter et al., 2017). The implementation of updated C<sub>ant</sub> estimation methods should additionally improve the accuracy and efficiency of both ESPERs and PyESPERs when C<sub>ant</sub> estimates are required. Future versions of ESPER written in MATLAB may be modified to improve interoperability with the Python implementation (i.e., to ensure the interpolation routines are identical in all instances between languages).

## 4 Data Availability

Data used for reconstruction and estimate comparisons is available through GLODAP (https://glodap.info; see Lauvset et al., 2022, doi:10.5194/essd-14-5543-2022 and Olsen et al., 2020, doi:10.5194/essd-12-3653-2020). The temperature and salinity gridded climatology created by Roemmich & Gilson (2009), doi:10.1016/j.pocean.2009.03.004 was created with data from the Argo Program.

## 5 Code Availability

PyESPERv1.0.0, affiliated files, and analyses files are available through LMD's GitHub page (https://github.com/LarissaMDias) and archived through Zenodo (doi: 10.5281/zenodo.15133085). Updates to

PyESPERv1.0.0 will also be published through LMD's GitHub page and archived through Zenodo. ESPERs (Carter, 2021) and original associated files used in creation of PyESPERv1.0.0 are available at BRC's GitHub page at <a href="https://github.com/BRCScienceProducts">https://github.com/BRCScienceProducts</a>. Input data used for comparisons are available through the GLODAP website (https://glodap.info).

## **6 Conclusions**

A near-replicate of ESPERs has been produced in the freely available Python programming language. This algorithm data product will allow Python users or researchers with limited funds an alternate, free method for using ESPERS (other than the proprietary MATLAB), increasing the accessibility of the original ESPER algorithms. The same logic applied to the original MATLAB ESPERs was applied within the Python coding language (PyESPERs, version 1.0.0), and results have demonstrated comparability to ESPER estimates. Estimates from PyESPER\_NNs precisely align with those from ESPER\_NNs for all equations and desired outcome variable combinations (Fig. 6) and estimates from these two routines align very closely for all estimates, and to within machine precision for all but pH<sub>T</sub> and DIC, which exhibit slight differences due to impacts of

interpolating for  $C_{\text{ant}}$ . PyESPER\_LIR estimates differ from ESPER\_LIR estimates for some coastal and deep-water regions between the two coding languages due to triangulation, extrapolation, and interpolation differences, but were more similar throughout all portions of the open ocean (Figs. 2, 3, and 4). Notably, PyESPER\_LIR performs equivalently to ESPER\_LIR when reconstructing the training data from GLODAPv2.v2022, so estimates produced from these two routines should be considered comparable rather than identical. Nevertheless, we do not recommend using PyESPER\_LIR in coastal or deep (>5500 m) waters when primarily interested in comparing results with those of the MATLAB implementation of ESPER\_LIR. Future updates to ESPERs will include updates to PyESPERs, with adjustments to allow for greater consistency and speed.

## 7 Appendices

390

395

## Appendix A: ESPER specifications

Sets of equations, predictor variables, and measurement uncertainties used in ESPER and PyESPER (adapted from Carter et al., 2021) are shown below.

Table A1: Input predictor variable combinations used for each ESPER equation (adapted from Carter et al., 2021), where S is salinity, T is temperature, and A, B, and C are defined in Table A2 (below).

| <b>Equation Number</b> | Predictor Variables |
|------------------------|---------------------|
| 1                      | S, T, A, B, C       |
| 2                      | S, T, A, C          |
| 3                      | S, T, B, C          |
| 4                      | S, T, C             |
| 5                      | S, T, A, B          |
| 6                      | S, T, A             |
| 7                      | S, T, B             |
| 8                      | S, T                |
| 9                      | S, A, B, C          |
| 10                     | S, A, C             |
| 11                     | S, B, C             |
| 12                     | S, C                |
| 13                     | S, A, B             |
| 14                     | S, A                |
| 15                     | S, B                |
| 16                     | S                   |

Table A2: Input predictor variables (A, B, and C) for each estimated property (adapted from Carter et al., 2021).

| <b>Estimated Property</b> | A         | В       | C        |
|---------------------------|-----------|---------|----------|
| TA                        | Nitrate   | Oxygen  | Silicate |
| DIC                       | Nitrate   | Oxygen  | Silicate |
| $pH_{\mathrm{T}}$         | Nitrate   | Oxygen  | Silicate |
| Phosphate                 | Nitrate   | Oxygen  | Silicate |
| Nitrate                   | Phosphate | Oxygen  | Silicate |
| Silicate                  | Phosphate | Oxygen  | Nitrate  |
| Oxygen                    | Phosphate | Nitrate | Silicate |

Table A3: Default measurement uncertainties ( $E_{Pi\_Default}$ ) for ESPERs and PyESPERs (adapted from Carter et al., 2021), where  $\theta$  is potential temperature.

| Units                 | Uncertainty                                               |
|-----------------------|-----------------------------------------------------------|
| _                     | 0.003, absolute                                           |
| $^{\circ}\mathrm{C}$  | 0.003, absolute                                           |
| μmol kg <sup>-1</sup> | 2%, relative                                              |
| μmol kg <sup>-1</sup> | 2%, relative                                              |
| μmol kg <sup>-1</sup> | 2%, relative                                              |
| μmol kg <sup>-1</sup> | 1%, relative                                              |
|                       | –<br>°C<br>μmol kg <sup>-1</sup><br>μmol kg <sup>-1</sup> |

Appendix B: Comparison using entire GLODAPv2.2022

Results of comparisons of PyESPER with ESPER for the entire GLODAPv2.2022 dataset, including the entire oceanic and coastal salinity range and data of all quality control flag categories are shown below.

Table B1: Mean (standard deviation), maximum, minimum, and normalized RMSE (RMSE<sub>n</sub>), for differences between MATLAB and Python LIRs, ESPER\_LIR and measured values, and PyESPER\_LIR and measured values for TA, DIC, pH<sub>T</sub>, phosphate, nitrate, silicate, and oxygen estimates (all units except pH<sub>T</sub> are μmol kg<sup>-1</sup>) for all equations combined, from the entire GLODAPv2.2022 (;; n=1,381,248).

|          |                         | Python - MATLAB       | MATLAB                |                       |                         | MATLAB.              | MATLAB - Measured     |                         |                        | Python - Measured    | Aeasured              |                       |
|----------|-------------------------|-----------------------|-----------------------|-----------------------|-------------------------|----------------------|-----------------------|-------------------------|------------------------|----------------------|-----------------------|-----------------------|
|          | Mean                    | Max                   | Min                   | $\mathbf{RMSE}_{n}$   | Mean                    | Max                  | Min                   | RMSEn                   | Mean                   | Max                  | Min                   | RMSEn                 |
| Ě        | -2.76x10 <sup>-2</sup>  | $6.46 \times 10^{2}$  | $-6.98x10^{2}$        | 1.12x10 <sup>-3</sup> | $3.36x10^{-1}$          | $1.19x10^3$          | $-7.34 \times 10^{2}$ | 5.12x10 <sup>-3</sup>   | $3.22x10^{-1}$         | $1.19x10^3$          | $-7.34$ x $10^2$      | 5.22x10 <sup>-3</sup> |
| ΗI       | (2.61)                  |                       |                       |                       | $(1.19x10^1)$           |                      |                       |                         | $(1.21x10^{1})$        |                      |                       |                       |
| Š        | $-5.35 \times 10^{-3}$  | $5.17x10^{2}$         | $-7.46 \times 10^{2}$ | $1.29 \times 10^{-3}$ | $-6.85 \times 10^{-1}$  | $9.79 \times 10^{2}$ | $-1.75 \times 10^{3}$ | 7.46x10 <sup>-3</sup>   | $-6.69 \times 10^{-1}$ | $9.79 \times 10^{2}$ | $-1.75x10^3$          | $7.45 \times 10^{-3}$ |
| DIC      | (2.82)                  |                       |                       |                       | $(1.63 \times 10^{1})$  |                      |                       |                         | $(1.63 \times 10^{1})$ |                      |                       |                       |
| i.       | -6.05x10 <sup>-5</sup>  | $7.65 \times 10^{-1}$ | $-1.59x10^{0}$        | $8.52 \times 10^{-4}$ | $2.03 \times 10^{-3}$   | $3.24x10^{0}$        | $-1.39 \times 10^{0}$ | 4.29x10 <sup>-3</sup>   | $2.14x10^{-3}$         | $2.85 \times 10^{0}$ | $-1.40x10^{0}$        | $4.26x10^{-3}$        |
| нd       | $(6.74 \times 10^{-3})$ |                       |                       |                       | $(3.39 \times 10^{-2})$ |                      |                       |                         | $(3.37x10^2)$          |                      |                       |                       |
| Phosp-   | $8.88x10^{-5}$          | 3.96                  | -3.31                 | $1.27 x 10^{-2}$      | -1.04x10 <sup>-3</sup>  | 4.77                 | $-1.37 \times 10^{1}$ | 5.74x10 <sup>-2</sup>   | -9.66x10⁴              | 4.66                 | $-1.39x10^{1}$        | $5.74 \times 10^{-2}$ |
| hate     | $(2.02 \times 10^{-2})$ |                       |                       |                       | $(9.26 \times 10^{-2})$ |                      |                       |                         | $(9.27x10^{-2})$       |                      |                       |                       |
| , , , ,  | $-7.07$ x $10^{-4}$     | $6.18x10^{1}$         | $-6.35x10^{1}$        | $1.99 \times 10^{-2}$ | -7.11x10 <sup>-3</sup>  | $8.17x10^{1}$        | $-1.82 \times 10^{2}$ | $5.70 \times 10^{-2}$   | $-7.79 \times 10^{-3}$ | $7.55 \times 10^{1}$ | $-1.76 \times 10^{2}$ | $5.76 \times 10^{-2}$ |
| Nitrate  | $(4.37x10^{-1})$        |                       |                       |                       | (1.27)                  |                      |                       |                         | (1.29)                 |                      |                       |                       |
| 0.11     | $3.78 \times 10^{-3}$   | $5.49 \text{x} 10^2$  | $-5.09x10^{2}$        | $4.39 \times 10^{-2}$ | $-1.22x10^{-1}$         | $3.11x10^{2}$        | $-1.11x10^3$          | $7.76x10^{-2}$          | $-1.16x10^{-1}$        | $1.16x10^{2}$        | $-5.61x10^{2}$        | $7.77 \times 10^{-2}$ |
| Sillcate | (2.24)                  |                       |                       |                       | (4.06)                  |                      |                       |                         | (4.03)                 |                      |                       |                       |
| Oxyge    | -4.47x10 <sup>-2</sup>  | $4.28\mathrm{x}10^2$  | $-5.78 \times 10^{2}$ | $1.58 \times 10^{-2}$ | $2.33x10^{-1}$          | $1.33x10^{3}$        | $-9.02 \times 10^{2}$ | $6.37 \text{x} 10^{-2}$ | $1.87x10^{-1}$         | $9.69x10^{2}$        | $-9.02x10^{2}$        | $6.19x10^{-2}$        |
| u        | (3.20)                  |                       |                       |                       | $(1.27x10^1)$           |                      |                       |                         | $(1.23 \times 10^{1})$ |                      |                       |                       |

MATLAB estimates from LIR

Figure B1: Difference between Python and MATLAB locally interpolated regression estimates (y-axis) compared to MATLAB estimates (x-axis) for whole ocean (w) data and all equations combined (22,099,968 total estimates from all equations for each variable), for TA (a), DIC (b), pH<sub>T</sub> (c), phosphate (d), nitrate (e), silicate (f), and oxygen (g) derived using all equations and calculated from entire GLODAPv2.2022 data (n=1,381,248). Units for all except pH<sub>T</sub> are in µmol kg<sup>-1</sup>. Top and bottom side histograms represent the distribution of the x and y axes, respectively. Note the differences in x- and y-axes scales. RMSEn is the normalized root mean square error, or the RMSE of all divided by the mean of all MATLAB estimates. The large range of sometimes unrealistic estimates along the x-axis can be attributed to anomalous and sometimes erroneous input data used for predictions.

Figure B2: Map of differences between Python and MATLAB ESPER locally interpolated regression estimates (total estimates n=22,099,968 for all variables) for the whole ocean ( $_w$ ), where small blue circles represent differences <2 x uncertainties of MATLAB estimates (n=22,034,967 for TA (a), 22,054,048 for DIC (b), 22,045,316 for pH<sub>T</sub> (c), 22,057,220 for phosphate (d), 22,045,770 for nitrate (e), 22,024,674 for silicate (f), and 22,045,827 for oxygen (g)), and red circles represent differences >2 x uncertainties of MATLAB estimates (n=65,001 for TA, 45,920 for DIC, 54,642 for pH, 42,748 for phosphate, 54,198 for nitrate, 75,294 for silicate, and 54,141 for oxygen; n=1,381,248).

Figure B3: Map of locations and depths (colorbar) where differences between Python and MATLAB ESPER locally interpolated regression estimates are greater than 2 x the estimate uncertainties for the whole ocean (w, n=22,034,967 for TA (a), 22,054,048 for DIC (b), 22,045,316 for pH<sub>T</sub> (c), 22,057,220 for phosphate (d), 22,045,770 for nitrate (e), 22,024,674 for silicate (f), and 22,045,827 for oxygen (g); n=1,381,248).

Python NNs, ESPER\_NN and measured values, and PyESPER\_NN and measured values for TA, DIC, pHT, phosphate, nitrate, silicate, and oxygen estimates (all units except pH<sub>T</sub> are μmol kg<sup>-1</sup>) for all equations combined, from the entire GLODAPv2.2022 dataset ("; where Table B2: Mean (standard deviation), maximum, minimum, and normalized RMSE (RMSE<sub>n</sub>), for differences between MATLAB and necessary input data were available, n=1,381,248).

|          |                         | Python - MATLAB         | AATLAB                 |                        | ř.                      | MATLAB.              | MATLAB - Measured     |                       |                         | Python - Measured    | Measured              |                       |
|----------|-------------------------|-------------------------|------------------------|------------------------|-------------------------|----------------------|-----------------------|-----------------------|-------------------------|----------------------|-----------------------|-----------------------|
|          | Mean                    | Max                     | Min                    | $\mathbf{RMSE}_{n}$    | Mean                    | Max                  | Min                   | RMSEn                 | Mean                    | Max                  | Min                   | RMSEn                 |
| ŧ        | -6.35x10 <sup>-12</sup> | $6.00 \times 10^{-6}$   | -9.00x10-6             | 2.69x10 <sup>-12</sup> | 4.99x10 <sup>-1</sup>   | $2.12x10^3$          | $-2.24x10^3$          | 6.30x10 <sup>-3</sup> | 4.99x10 <sup>-1</sup>   | $2.12x10^3$          | $-2.24 \times 10^3$   | $6.30 \times 10^{-3}$ |
| IA       | $(6.24x10^{-9})$        |                         |                        |                        | $(1.46 \times 10^{1})$  |                      |                       |                       | $(1.46 \times 10^{1})$  |                      |                       |                       |
| Ç        | $-3.24$ x $10^{-3}$     | 2.88                    | -4.68                  | 5.19x10 <sup>-5</sup>  | $-4.82x10^{-1}$         | $1.97x10^3$          | $-2.22x10^3$          | $8.01x10^{-3}$        | $-4.82 \times 10^{-1}$  | $1.97x10^3$          | $-2.22x10^3$          | $8.01x10^{-3}$        |
| DIC      | $(1.13x10^{-1})$        |                         |                        |                        | $(1.75 \times 10^{1})$  |                      |                       |                       | $(1.75 \times 10^{1})$  |                      |                       |                       |
| 11       | $6.08x10^{-6}$          | $1.21x10^{-2}$          | $-2.03x10^{-2}$        | 4.52x10 <sup>-5</sup>  | $-3.01$ x $10^{-3}$     | 2.53                 | -5.74                 | 4.49x10 <sup>-3</sup> | $-3.00 \times 10^{-3}$  | 2.53                 | -5.74                 | 4.49x10 <sup>-3</sup> |
| рпт      | $(3.58x10^{-4})$        |                         |                        |                        | $(3.54x10^{-3})$        |                      |                       |                       | $(3.54x10^{-2})$        |                      |                       |                       |
| 2        | $6.32 \times 10^{-14}$  | $6.39x10^{-7}$          | $-1.25x10^{-7}$        | $1.31x10^{-10}$        | $-5.84x10^{-4}$         | $1.14x10^{1}$        | -6.02                 | $5.06 \times 10^{-2}$ | $-5.84 \times 10^{-4}$  | $1.14x10^{1}$        | -6.02                 | $5.06 \times 10^{-2}$ |
| -dsou-   | $(2.08x10^{-}$          |                         |                        |                        | $(8.25 \times 10^{-2})$ |                      |                       |                       | $(8.25 \times 10^{-2})$ |                      |                       |                       |
| nate     | (01                     |                         |                        |                        |                         |                      |                       |                       |                         |                      |                       |                       |
| 7.14     | $5.12x10^{-13}$         | $1.17x10^{-5}$          | -2.28x10 <sup>-6</sup> | $1.39 x 10^{-10}$      | $-1.07$ x $10^{-2}$     | $1.97x10^{2}$        | $-1.45 \times 10^{2}$ | 5.06x10 <sup>-2</sup> | $-1.07 \times 10^{-2}$  | $1.97x10^{2}$        | $-1.45 \times 10^{2}$ | $5.06 \times 10^{-2}$ |
| nitrate  | $(3.06x10^{-9})$        |                         |                        |                        | (1.17)                  |                      |                       |                       | (1.17)                  |                      |                       |                       |
| :        | $-2.35x10^{-13}$        | $7.46x10^{-6}$          | -2.97x10 <sup>-6</sup> | $4.96x10^{-11}$        | $-2.37$ x $10^{-2}$     | $6.25 \times 10^{2}$ | $-7.32 \times 10^{2}$ | $7.06x10^{-2}$        | $-2.37$ x $10^{-2}$     | $6.25 \times 10^{2}$ | $-7.32 \times 10^{2}$ | $7.06 \times 10^{-2}$ |
| Silicate | $(2.55 \times 10^{-9})$ |                         |                        |                        | (3.71)                  |                      |                       |                       | (3.71)                  |                      |                       |                       |
|          | $-4.65x10^{-13}$        | $1.00 \text{x} 10^{-9}$ | $-1.00x10^{-7}$        | $1.06 x 10^{-12}$      | $-3.46 \times 10^{-3}$  | $7.12x10^{2}$        | $-1.22x10^{3}$        | 5.65x10 <sup>-2</sup> | $-3.46 \times 10^{-2}$  | $7.12x10^2$          | $-1.22x10^3$          | $5.65 \times 10^{-2}$ |
| CAYBC    | $(2.15x10^{-}$          |                         |                        |                        | $(1.13 \times 10^1)$    |                      |                       |                       | $(1.13 \times 10^{1})$  |                      |                       |                       |
| п        | 10)                     |                         |                        |                        |                         |                      |                       |                       |                         |                      |                       |                       |

MATLAB estimates from NN

Figure B4: Difference between Python and MATLAB neural network estimates (*y*-axis) compared to MATLAB estimates (*x*-axis) for whole ocean (*w*) data and all equations combined for TA (a, 17,802,134 total estimates from all equations), DIC (b, 17,802,134 estimates), pH<sub>T</sub> (c, 17,799,566 estimates), phosphate (d, 17,802,134 estimates), nitrate (e, 17,395,954 estimates), silicate (f, 17,445,310 estimates), and oxygen (g, 17,220,360 estimates) derived using all equations and calculated from entire GLODAPv2.2022 dataset (*n*=1,381,248). Units for all except pH<sub>T</sub> are in μmol kg<sup>-1</sup>. Top and bottom side histograms represent the distribution of the x and y axes, respectively. Note the differences in *x*- and *y*-axes scales. RMSE<sub>n</sub> is the normalized root mean square error, or the RMSE of all divided by the mean of all estimates. The large range of sometimes unrealistic estimates along the x-axis can be attributed to anomalous and sometimes erroneous input data used for predictions.

## Appendix C: Example of mapped DIC estimates from PyESPER and ESPER

Surface ocean DIC estimates form PyESPER\_LIR and ESPER\_LIR applied to the Roemmich and Gilson climatology (Roemmich and Gilson, 2009). Differences in surface ocean DIC between the two coding languages (c) illustrate the need to avoid using PyESPER\_LIR for DIC in the surface ocean when comparing to MATLAB ESPER\_LIR.

465

Figure C1: Maps of 2023 mean annual surface estimates of MATLAB ESPER\_LIR DIC (a), Python PyESPER\_LIR DIC (b), and PyESPER\_LIR – ESPER\_LIR DIC (c; units are μmol kg<sup>-1</sup>) from application of ESPERs to the Roemmich and Gilson Argo-based (Argo, 2000) climatology (Roemmich and Gilson, 2009).

# 470 Appendix D: Comparison of interpolation and extrapolation values between MATLAB and Python

MATLAB ESPER\_LIRs avoid extrapolation by addition of a false set of data points at very far distances from the grid. However, when this method was implemented in Python, significant errors were introduced due to the differences in triangulation (which were both valid mathematical solutions) between coding languages. Therefore, it was necessary to find another means of calculating extrapolations in PyESPER\_LIRs which was more similar to those of ESPER\_LIRs. We did this by producing a larger grid in MATLAB and reading that into Python. A simple demonstration of the errors introduced by this method is described below.

For this comparison we imagine a hypothetical cube, with x, y, and z coordinates, upon which we wish to provide estimates for a fourth variable (p) via both interpolation and extrapolation (Fig. D1a). We have created a random dataset of points and values within this cube for these demonstration purposes. We then followed the same procedure as in the PyESPER data product creation, whereby we extended this grid in three-dimensional space and used MATLAB scatteredInterpolant extrapolations to estimate values on the expanded grid (Fig. D1b). This method conducts a Delaunay triangulation, then uses both linear interpolation and extrapolation to estimate values. These extrapolated values were then used for interpolation only within Python using scipy's Delaunay and LinearNDInterpolator functions, which produced more consistent results than interpolation and extrapolation within Python.

Figure D1: Hypothetical "grid" whereby estimates (p) interpolated within the grid are shown in blue and extrapolations are shown in red (a). Grid created for demonstration purposes, with interpolated values in blue and areas where we extrapolated values in red (b).

When interpolations within Python were compared to locations on the hypothetical grid where interpolations occurred in MATLAB also, results were more similar than those where the grid was extrapolated within MATLAB. This is because

different, but equally valid, mathematics are used to interpolate and extrapolate. Namely, a triangulation is used as the basis for interpolations, whereas extrapolations are based on boundary gradients. Despite these differences, results were still more similar with this method between the two coding languages than when extrapolations were done in both Python and MATLAB.

Table D1: Comparison of differences between MATLAB interpolations and extrapolations and Python results (all interpolations).

|                    | MATLAB Interpolation - Python | MATLAB Extrapolation - Python |
|--------------------|-------------------------------|-------------------------------|
|                    | Interpolation                 | Interpolation                 |
| Mean               | 0.0004                        | -0.6693                       |
| Standard Deviation | 0.9559                        | 5.2088                        |
| Max                | 2.2582                        | 13.3083                       |
| Min                | -2.4593                       | -15.6633                      |

## **8 Author Contributions**

LMD was primarily responsible for Python data product development, validation, formal analysis, investigation, data curation, writing, and visualization. BRC primarily responsible for project conceptualization, MATLAB data product development, supervision, project administration, providing resources, funding acquisition, and editing. Methods were devised by both LMD and BRC.

## **9 Competing Interests**

The authors declare that they have no conflict of interest.

#### 10 Acknowledgments

The University of Washington Cooperative Institute for Climate, Ocean, and Ecosystem Studies (CICOES) has assigned CICOES Publication Contribution Number 2024-1382. The National Oceanic and Atmospheric Administration (NOAA) Pacific Marine Environmental Laboratory has assigned PMEL Contribution Number 5707. BRC and LMD thank the OAR Climate Program Office and NOAA's Global Ocean Monitoring and Observation program for support under award number NA21OAR4310251. The data used for DIC data products were collected and made freely available by the International Argo Program and the national programs that contribute to it (http://www.argo.ucsd.edu, http://argo.jcommops.org). The Argo Program is part of the Global Ocean Observing System. BRC and LMD also would like to sincerely thank Matthew

Humphreys, who served not only as a reviewer but also provided careful editing and help with packaging the PyESPER code. BRC and LMD also thank Daniel Sandborn, who provided useful Python coding tips.

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
