# Peer review of "PyESPERv1.0.0: A Python implementation of empirical seawater property estimation routines (ESPERs)"

_EGUsphere, 2025_

## Referee Comment (RC1)

Overall great. The first iteration of this in Matlab was already sound in my opinion so this translation requires less scrutiny. I have not run the code myself, and although it would be intensive, I believe the accessibility would improve significantly if there is a possibility for a computer scientist to create a simple UI for either packages.

40 - Should add note of the potential high error when using a model to estimate a variable then used to calculate carbonate chemistry parameter without nutrient information too

50 - I would argue that it may not be considered entirely findable for many scientists who are not coding competent and even those who are, are likely unaware of the Zenodo and GitHub repositories though I recognize that is not entirely your responsibility

68 - If all models perform comparably then why is there a need for all three why not just use the mixed as an ensemble prediction

100 - if there's inadequate data number and the area size is doubled, does the output indicate this? Has it been checked if this correlates with an increase in error? Why is it jumping straight to double instead of small increase intervals?

160/172 - Should add a caveat that in addition to not predicting past 2030 they should not be used in areas with abnormal atmospheric $CO^2$ absorption or profiles ie. upwelling, coastal areas, high freshwater outflow mentioned in 261 and may seem obvious to some but not others

---

## Referee Comment (RC4)

**Review of "PyESPERv1.01.01: A Python implementation of empirical seawater property estimation routines (ESPERs)" by L.M. Dias and B.R. Carter**

28th May 2025

**Overview**

[1] This manuscript presents a Python implementation of an existing MATLAB tool to estimate values for various marine carbonate system and nutrient parameters in seawater. There is no new development of the tool here. The aim is a direct translation. This is a valuable goal, because unlike MATLAB, Python is free and open source. But it does mean that, in my opinion, the quality of the code itself is equally as important as (or even more important than) the manuscript when assessing this submission. The manuscript is primarily describing the new code, so the new code must be complete before publication. I do not think that the code in its current form is complete, usable and publishable, but I think it is possible for this to be achieved within the scope of revisions to this submission (manuscript and code).

[2] My sense from looking through the other reviews is that they are mostly focused on the manuscript rather than the code, so this review deliberately focuses mostly on the code rather than the manuscript.

[3] Although my comments may read as being rather critical, they are all intended to be constructive and I am overall really positive about this manuscript and (more importantly) the code. I'm very happy to see it appear in Python and it's certainly something that I could see myself using in the future. Thank you to the authors for their efforts!

**MATLAB-Python differences**

[4] The authors acknowledge that Python code does not produce exactly the same results as the MATLAB. They argue that this is mostly due to differences in how Delaunay triangulation and extrapolation are implemented by the external packages used to do these steps. This argument is plausible but it is not yet convincing. Is there some way the authors can prove that this is the cause of the differences, or at least demonstrate it more quantitatively? See also [8] below.

[5] Continuing on the above, I am worried about the word "most" ("this difference in implementation is the source of most disagreements"; line 87). This implies that there remain some differences that cannot be explained in this way, which presumably points to bugs in the code? See also [9] below.

[6] The test dataset does include some additional cruises that were not part of the training set but it is not really independent. The additional cruises will have been assessed for consistency with the existing GLODAP product and potentially had their values adjusted to match better.

[7] If I understood Section 3.1.1 correctly (especially lines 260-264), the 'extrapolation' areas generally had bigger differences than the 'interpolation' areas. This is puzzling. My

understanding from Section 2.1.1 (lines 110-128) was that the Python implementation does not extrapolate itself, but rather reads a from saved output for the extrapolation regions generated by the MATLAB implementation. If that's right, then surely these regions should agree very well with each other, because Python is just copying MATLAB directly rather than doing the calculations internally? Perhaps I have misunderstood the explanations – in which case the corresponding text should be made clearer.

[8] From Figure 2, some of the differences are really rather large (e.g. up to 200 µmol/kg in DIC, 0.5 in pH). Without further evidence I find it hard to understand how or accept that such a large difference could really be due to differences in how Delaunay triangles are calculated. A clearer explanation of this would be appreciated.

[9] Were this being released as a data product, then the issues above would be less important, because of the validation against the GLODAP dataset for example. However, this is a tool intended for users to calculate things with untested sets of input conditions. If some part of the differences between implementations are due to bugs in the code, they cannot be written off just because they're fairly small in these tests, because they could easily have a much bigger effect with a different set of inputs. In order to have confidence in the results, any unexpected behaviour or differences between implementations above the level of computer precision must be really thoroughly understood.

**Code quality**

[10] I was able to get the example code to run but it still required some troubleshooting and corrections to the code beyond the instructions given in the README. These were mostly related to defining and concatenating file paths (which can more robustly and conveniently be done with os.path.join rather than by manually manipulating strings). I have made a pull request (PR) to the GitHub repo which contains these and some other (see [11]) fixes (https://github.com/LarissaMDias/PyESPER/pull/1).

[11] Parts of the code are very difficult to follow. This makes me worry more about points [5] and [9] above. The most critical issues are:

- The functions needed are in a Jupyter notebook, so they can't be imported and used in other workflows.
- There are two notebooks both with copies of these functions – there should only be one "source of truth".
- Variables are defined, renamed and copied without clear reasons why, making it easy to lose track of which version of a variable should be used for the next step of the calculation.
- The deprecated seawater package is used instead of its well-maintained successor gsw.
- It's virtually never necessary to explicitly use global variables in Python and best practice to avoid doing so.
- Numerical data appear to be processed into strings at some points?

Some more minor points that would improve things:

- Variables are converted between dicts and pandas DataFrames, and lists and numpy arrays, often without any clear reason. Both for code clarity and

computational speed, numerical data should be kept as numpy arrays throughout, and dicts promoted to DataFrames only when essential.
- Some packages are imported and not used (e.g., decimal).
- Some variables are defined and never used.
- Sometimes multiple packages are used where one would be more efficient (e.g., using math and statistics for some calculations that should all be done with numpy).
- The code could be run through a linter / auto-styler (e.g. Ruff, Black) to make it more readable and help locate some of the issues noted above.

The PR I made to the GitHub repo (see [10]) also contains fixes for some, but not all, of the points above, and I'd be happy to discuss with the authors further on how to tackle any of these issues if that might be useful.

[12] Following from [10], the authors note that the Python code runs significantly slower than the MATLAB. I suspect the frequent reliance on looping calculations through lists, which is known to be very slow in Python, rather than vectorising calculations across numpy arrays, may be largely responsible for this. Operations on pandas DataFrames can also be a lot slower than the equivalent with a dict or numpy array.

[13] For this to be really considered "available" in Python it needs at the very least to be packaged properly and installable from the GitHub repo with pip. Functions in Jupyter notebooks are not useful for integrating into other workflows. Given my comments in [1], that this manuscript is really about the code, I think that should be a bare minimum for publication.

[14] Uploading to PyPI and conda-forge would be very useful additional steps, although not critical for publishing this manuscript.

**Minor comments**

[15] Figure 2: the y-axis scales have very unusual intervals, which does make it harder to interpret the figures.

[16] Line 261-262: presumably "these locations" refers to the "exceptions" from the previous sentence rather than the "most ocean regions", but this is not clear.

[17] The version number 1.01.01 is quite unusual. Of course it's the authors' prerogative to use whatever system they like, but I would suggest considering switching to the very widely used semantic versioning (https://semver.org) to make it easier to interpret.

[18] For the examples, you could consider using https://github.com/mvdh7/glodap to import the GLODAP dataset (this automatically downloads the files if the user doesn't have them). I included an example script in my PR (see [10]) which shows how this could be implemented.

---

## Author Comment (AC4)

**Review of "PyESPERv1.01.01: A Python implementation of empirical seawater property estimation routines (ESPERs)" by L.M. Dias and B.R. Carter**

*28th May 2025*
*Note: Also see Git commits and notes from Matthew email*

**Overview**

*[1] This manuscript presents a Python implementation of an existing MATLAB tool to estimate values for various marine carbonate system and nutrient parameters in seawater. There is no new development of the tool here. The aim is a direct translation. This is a valuable goal, because unlike MATLAB, Python is free and open source. But it does mean that, in my opinion, the quality of the code itself is equally as important as (or even more important than) the manuscript when assessing this submission. The manuscript is primarily describing the new code, so the new code must be complete before publication. I do not think that the code in its current form is complete, usable and publishable, but I think it is possible for this to be achieved within the scope of revisions to this submission (manuscript and code).*

**We thank you for the constructive comment and agree that the code can be greatly improved quickly with a few minor fixes prior to publication, especially with your advice below and on GitHub.**

*[2] My sense from looking through the other reviews is that they are mostly focused on the manuscript rather than the code, so this review deliberately focuses mostly on the code rather than the manuscript.*

**We appreciate the coding expertise.**

*[3] Although my comments may read as being rather critical, they are all intended to be constructive and I am overall really positive about this manuscript and (more importantly) the code. I'm very happy to see it appear in Python and it's certainly something that I could see myself using in the future. Thank you to the authors for their eKorts!*

**We thank you for the constructive comment.**

**MATLAB-Python di5erences**

*[4] The authors acknowledge that Python code does not produce exactly the same results as the MATLAB. They argue that this is mostly due to diKerences in how Delaunay triangulation and extrapolation are implemented by the external packages used to do these steps. This argument is plausible but it is not yet convincing. Is there some way the authors can prove that this is the cause of the diKerences, or at least demonstrate it more quantitatively? See also [8] below.*

**We have adjusted the explanation as follows and added an appendix (D) to the manuscript that provides an improved and more detailed quantitative explanation and comparison of this issue.**

**L327-332. Spatial patterns in distribution of outliers shown in Fig. 4 appear to reflect locations where more edge-of-grid biogeochemical measurements were collected (e.g., near coasts and in deep waters). Hence, these locations aligned well with places where coefficients were extrapolated in MATLAB for use in PyESPER_LIRs, compared to interpolations with far away "dummy points" within MATLAB ESPER_LIRs (see Sect. 2.1.1, "*Locally interpolated regressions*"; Figs. 3, 4, and 5; for w Fig. B2 and B3). Within regions where MATLAB and Python were interpolating similarly, far outliers were uncommon (Figs. 3, 4, 5, B2, and B3).**

*[5] Continuing on the above, I am worried about the word "most" ("this diKerence in implementation is the source of most disagreements"; line 87). This implies that there remain some diKerences that cannot be explained in this way, which presumably points to bugs in the code? See also [9] below.*

**This choice of wording was indeed misleading, as we believe the interpolation differences beyond machine precision to be entirely due to interpolation differences. See the corrected wording below (simply omitting the word "most"):**

**L107-109. The three-dimensional interpolation algorithm is implemented differently in MATLAB and Python, and although both calculations are valid, this difference in implementation is the source of disagreements we find and later quantify between ESPER and PyESPER.**

*[6] The test dataset does include some additional cruises that were not part of the training set but it is not really independent. The additional cruises will have been assessed for consistency with the existing GLODAP product and potentially had their values adjusted to match better.*

**This is true. However, the GLODAP dataset was also used to validate ESPERs, which was our rationale for choosing this dataset. In future updates, we plan to also validate both ESPERs and PyESPERs against other datasets and potentially model results in an independent analysis.**

*[7] If I understood Section 3.1.1 correctly (especially lines 260-264), the 'extrapolation' areas generally had bigger diKerences than the 'interpolation' areas. This is puzzling. My*
2
*understanding from Section 2.1.1 (lines 110-128) was that the Python implementation does not extrapolate itself, but rather reads a from saved output for the extrapolation regions generated by the MATLAB implementation. If that's right, then surely these regions should agree very well with each other, because Python is just copying MATLAB directly rather than doing the calculations internally? Perhaps I have misunderstood the explanations – in which case the corresponding text should be made clearer.*

**This is a good point and a complicated issue, but it is important to note that Python is not copying MATLAB directly. We have added more information about what the two ESPER versions are doing below and in Appendix D.**

**MATLAB ESPER_LIR: The grid is expanded vastly (to very large numbers) in order to avoid extrapolation.**

**Python PyESPER_LIR: The above method resulted in extremely different values due to different triangulation methods in Python. Instead, we extrapolated the grid within MATLAB and used this larger, extrapolated grid to interpolate within Python. After extensive testing of many methods, this was the closest agreement method possible.**

**Please note that in updates we hope to find interpolation methods that match precisely between MATLAB and Python.**

*[8] From Figure 2, some of the diKerences are really rather large (e.g. up to 200 μmol/kg in DIC, 0.5 in pH). Without further evidence I find it hard to understand how or accept that such a large diKerence could really be due to diKerences in how Delaunay triangles are calculated. A clearer explanation of this would be appreciated.*

**Please see the above comments and addition of Appendix D and the table of differences below for a randomly created variable with values between 1-10. Because neural networks agreed to within machine precision, and we have noted these differences between interpolation for the two languages, we can conclude that indeed the interpolation methods introduced the differences.**

**Table D1: Comparison of differences between MATLAB interpolations and extrapolations and Python results (all interpolations).**

|  | MATLAB Interpolation - Python Interpolation | MATLAB Extrapolation - Python Interpolation |
|---|---|---|
| **Mean** | 0.0004 | -0.6693 |
| **Standard Deviation** | 0.9559 | 5.2088 |
| **Max** | 2.2582 | 13.3083 |
| **Min** | -2.4593 | -15.6633 |

*[9] Were this being released as a data product, then the issues above would be less important, because of the validation against the GLODAP dataset for example. However, this is a tool intended for users to calculate things with untested sets of input conditions. If some part of the diKerences between implementations are due to bugs in the code, they cannot be written oK just because they're fairly small in these tests, because they could easily have a much bigger eKect with a diKerent set of inputs. In order to have confidence in the results, any unexpected behaviour or diKerences between implementations above the level of computer precision must be really thoroughly understood.*

**This is a valid point, which we believe we have addressed through addition of Appendix D.**

**Code quality**

**Please note that this review is due in the next few days, but we plan on more thoroughly addressing all of these code issues in the next week. Please do not hesitate to email us if you wish to check in!**

*[10] I was able to get the example code to run but it still required some troubleshooting and corrections to the code beyond the instructions given in the README. These were mostly related to defining and concatenating file paths (which can more robustly and conveniently be done with os.path.join rather than by manually manipulating strings). I have made a pull request (PR) to the GitHub repo which contains these and some other (see [11]) fixes (https://github.com/LarissaMDias/PyESPER/pull/1).*

**Thank you for the useful comments. We have accepted and merged this pull request.**

*[11] Parts of the code are very diKicult to follow. This makes me worry more about points [5] and [9] above. The most critical issues are:*

**We agree that (as marine chemists) we have no formal training in coding and the code may be sloppy. We thank you for your careful edits!**

*• The functions needed are in a Jupyter notebook, so they can't be imported and used in other workflows.*

**We now have .py modules available and are near-completion of packaging. We have also completely eliminated the JupyterNotebooks from the repository.**

*• There are two notebooks both with copies of these functions – there should only be one "source of truth".*

**We now have only one copy of each function.**

*• Variables are defined, renamed and copied without clear reasons why, making it easy to lose track of which version of a variable should be used for the next step of the calculation.*

**We are working on close editing of code and variables within, for a more streamlined code in the final version.**

*• The deprecated seawater package is used instead of its well-maintained successor gsw.*

**This is done for consistency with the current MATLAB version, but will be changed to the gsw package for future updates. We will also work on including an option in this version for users to go ahead and use the gsw package for users who do not wish to compare results to current MATLAB versions.**

• *It's virtually never necessary to explicitly use global variables in Python and best practice to avoid doing so.*

**Thank you, we will remove unnecessary global variables.**

• *Numerical data appear to be processed into strings at some points?*

**Some of the functions used required string formatting; however, we will look into whether there is an improved solution for this that does not require string formatting.**

*Some more minor points that would improve things:*
• *Variables are converted between dicts and pandas DataFrames, and lists and numpy arrays, often without any clear reason. Both for code clarity and*
*3*
*computational speed, numerical data should be kept as numpy arrays throughout, and dicts promoted to DataFrames only when essential.*

**Thank you for these tips. We will indeed reformat to this recommendation.**

• *Some packages are imported and not used (e.g., decimal).*

**We will eliminate these.**

• *Some variables are defined and never used.*

**This is odd, but a thorough comb-through of the code will help.**

• *Sometimes multiple packages are used where one would be more eKicient (e.g., using math and statistics for some calculations that should all be done with numpy).*

**Thank you for the advice. We will try to use fewer packages for our calculations in the final version.**

• *The code could be run through a linter / auto-styler (e.g. RuK, Black) to make it more readable and help locate some of the issues noted above.*

**This is a good idea, and we will execute this once we have finished all preliminary editing.**

*The PR I made to the GitHub repo (see [10]) also contains fixes for some, but not all, of the points above, and I'd be happy to discuss with the authors further on how to tackle*

*any of these issues if that might be useful.*

**We thank you and will work with you to make this more user-friendly. Even though we are still working on making these changes, please feel free to check up through email.**

*[12] Following from [10], the authors note that the Python code runs significantly slower than the MATLAB. I suspect the frequent reliance on looping calculations through lists, which is known to be very slow in Python, rather than vectorising calculations across numpy arrays, may be largely responsible for this. Operations on pandas DataFrames can also be a lot slower than the equivalent with a dict or numpy array.*

**We have rewritten this section and table; most of our issues stemmed from using JupyterNotebooks. However, we will consider your above comments for even greater speed.**

*[13] For this to be really considered "available" in Python it needs at the very least to be packaged properly and installable from the GitHub repo with pip. Functions in Jupyter notebooks are not useful for integrating into other workflows. Given my comments in [1], that this manuscript is really about the code, I think that should be a bare minimum for publication.*

**We thank you and will make it installable with pip once code editing is finished.**

*[14] Uploading to PyPI and conda-forge would be very useful additional steps, although not critical for publishing this manuscript.*

**We agree and will also plan on doing this, with less time constraint than the aforementioned recommendations.**

**Minor comments**

*[15] Figure 2: the y-axis scales have very unusual intervals, which does make it harder to interpret the figures.*

**We have changed the y-axis scales of Figure 2 to be much more readable, and whole-number intervals when possible.**

*[16] Line 261-262: presumably "these locations" refers to the "exceptions" from the previous sentence rather than the "most ocean regions", but this is not clear.*

**We have altered the language to "these exceptionally different locations"**

*[17] The version number 1.01.01 is quite unusual. Of course it's the authors' prerogative to use whatever system they like, but I would suggest considering switching to the very widely used semantic versioning (https://semver.org) to make it easier to interpret.*

**If we understood correctly, all version numbers for this initial release should (and have been) altered to 1.0.0.**

*[18] For the examples, you could consider using https://github.com/mvdh7/glodap to import the GLODAP dataset (this automatically downloads the files if the user doesn't have them). I included an example script in my PR (see [10]) which shows how this could be implemented.*

**We thank you for the information and have included this method in our examples, rather than prior downloaded datasets.**

---

## Author Response (AR2)

**Final Response: Andrew Yool**

Thank you for your revised manuscript and response to your (many!) referees.

**We thank you for the opportunity to greatly improve the manuscript and code.**

I have reviewed these now and am generally satisfied that your revisions and expansions satisfy the majority of the issues raised by your referees. However, I note - particularly in your responses to referee #4 - that your replies appear provisional with reference to planned or anticipated revisions to the model code. While this may just be a colloquial language in your response, it does suggest that the code described by this manuscript is still undergoing changes. While it is to be expected that the code will undergo revision in the future, it is important that this manuscript describes a specific instance (and fixed version number) of the code that is satisfactory to your referees.

**We apologize for lack of clarity and have finished all code revisions that were required for this version.**

To which end, I would appreciate it if you could clarify your responses to your referees as necessary so that it is absolutely clear where work has been concluded on this version. Where future (even imminent) change is expected, this should be noted, but your manuscript should absolutely be clear on where the final accepted version of your code ends and where "future development" is anticipated. The current "still working on this" tone of some of your responses is not acceptable where a very specific model version is being described and formalised.

**We have clarified the responses to Reviewer #4 and attached those to the end of this document. Specifically, we have addressed all immediate issues and have finished with the current code version.**

As such, I am returning your manuscript and response to you for what I expect will be a final iteration. The positivity of your referees means I do not anticipate the need to return your manuscript to them. If you require any additional time to satisfy this request, please do not hesitate to get in contact with me - I appreciate that manuscript revision is in competition with your other research activities (and the upcoming summer leave period).

**We thank you for your consideration.**

\* One further point: Figure B3 uses a rainbow palette that is incompatible with our requirements around colour blindness. Could you please revise this to one of the other palettes that you use that is consistent with this requirement?

We apologize for forgetting to do the colorblind correction of this figure and have updated it to adhere to the guidelines and match Fig. 4. Please see L605 and the improved figure B3 below.

**Additional (Edited) Responses to Reviewer #4 (MH):**

Review of "PyESPERv1.01.01: A Python implementation of empirical seawater property estimation routines (ESPERs)" by L.M. Dias and B.R. Carter 28th May 2025

Note: Also see Git commits and notes from Matthew email Overview

[1] This manuscript presents a Python implementation of an existing MATLAB tool to estimate values for various marine carbonate system and nutrient parameters in

seawater. There is no new development of the tool here. The aim is a direct translation. This is a valuable goal, because unlike MATLAB, Python is free and open source. But it does mean that, in my opinion, the quality of the code itself is equally as important as (or even more important than) the manuscript when assessing this submission. The manuscript is primarily describing the new code, so the new code must be complete before publication. I do not think that the code in its current form is complete, usable and publishable, but I think it is possible for this to be achieved within the scope of revisions to this submission (manuscript and code).

We thank you for the constructive comment and agree that the code can be greatly improved quickly with a few minor fixes prior to publication, especially with your advice below and on GitHub.

[2] My sense from looking through the other reviews is that they are mostly focused on the manuscript rather than the code, so this review deliberately focuses mostly on the code rather than the manuscript.

**We appreciate the coding expertise.**

[3] Although my comments may read as being rather critical, they are all intended to be constructive and I am overall really positive about this manuscript and (more importantly) the code. I'm very happy to see it appear in Python and it's certainly something that I could see myself using in the future. Thank you to the authors for their efforts!

**We thank you for the constructive comment.**

**MATLAB-Python differences**

[4] The authors acknowledge that Python code does not produce exactly the same results as the MATLAB. They argue that this is mostly due to differences in how Delaunay triangulation and extrapolation are implemented by the external packages used to do these steps. This argument is plausible but it is not yet convincing. Is there some way the authors can prove that this is the cause of the differences, or at least demonstrate it more quantitatively? See also [8] below.

We have adjusted the explanation as follows and added an appendix (D) to the manuscript that provides an improved and more detailed quantitative explanation and comparison of this issue.

L327-332. Spatial patterns in distribution of outliers shown in Fig. 4 appear to reflect locations where more edge-of-grid biogeochemical measurements were collected (e.g., near coasts and in deep waters). Hence, these locations aligned well with places where coefficients were extrapolated in MATLAB for use in PyESPER\_LIRs, compared to interpolations with far away "dummy points" within MATLAB ESPER\_LIRs (see Sect. 2.1.1, "Locally interpolated regressions"; Figs. 3, 4, and 5; for w Fig. B2 and B3). Within regions where MATLAB and Python were interpolating similarly, far outliers were uncommon (Figs. 3, 4, 5, B2, and B3).

[5] Continuing on the above, I am worried about the word "most" ("this difference in implementation is the source of most disagreements"; line 87). This implies that there remain some differences that cannot be explained in this way, which presumably points to bugs in the code? See also [9] below.

This choice of wording was indeed misleading, as we believe the interpolation differences beyond machine precision to be entirely due to interpolation differences. See the corrected wording below (simply omitting the word "most"):

L107-109. The three-dimensional interpolation algorithm is implemented differently in MATLAB and Python, and although both calculations are valid, this difference in implementation is the source of disagreements we find and later quantify between ESPER and PyESPER.

[6] The test dataset does include some additional cruises that were not part of the training set but it is not really independent. The additional cruises will have been assessed for consistency with the existing GLODAP product and potentially had their values adjusted to match better.

This is true. However, the GLODAP dataset was also used to validate ESPERs, which was our rationale for choosing this dataset. In future updates, we plan to also validate both ESPERs and PyESPERs against other datasets and potentially model results in an independent analysis.

[7] If I understood Section 3.1.1 correctly (especially lines 260-264), the 'extrapolation' areas generally had bigger differences than the 'interpolation' areas. This is puzzling. My understanding from Section 2.1.1 (lines 110-128) was that the Python implementation does not extrapolate itself, but rather reads a from saved output for the extrapolation regions generated by the MATLAB implementation. If that's right, then surely these regions should agree very well with each other, because Python is just copying MATLAB directly rather than doing the calculations internally? Perhaps I have misunderstood the explanations – in which case the corresponding text should be made clearer.

This is a good point and a complicated issue, but it is important to note that Python is not copying MATLAB directly. We have added more information about what the two ESPER versions are doing below and in Appendix D.

MATLAB ESPER\_LIR: The grid is expanded vastly (to very large numbers) in order to avoid extrapolation.

Python PyESPER\_LIR: The above method resulted in extremely different values due to different triangulation methods in Python. Instead, we extrapolated the grid within MATLAB and used this larger, extrapolated grid to interpolate within Python. After extensive testing of many methods, this was the closest agreement method possible.

Please note that in updates we hope to find interpolation methods that match precisely between MATLAB and Python.

[8] From Figure 2, some of the differences are really rather large (e.g. up to  $200 \mu mol/kg$  in DIC, 0.5 in pH). Without further evidence I find it hard to understand how or accept that such a large difference could really be due to differences in how Delaunay triangles are calculated. A clearer explanation of this would be appreciated.

Please see the above comments and addition of Appendix D and the table of differences below for a randomly created variable with values between 1-10. Because neural networks agreed to within machine precision, and we have noted these differences between interpolation for the two languages, we can conclude that indeed the interpolation methods introduced the differences.

Table D1: Comparison of differences between MATLAB interpolations and extrapolations and Python results (all interpolations).

|                           | MATLAB Interpolation -
Python Interpolation | MATLAB Extrapolation -
Python Interpolation |
|---------------------------|------------------------------------------------|------------------------------------------------|
| Mean                      | 0.0004                                         | -0.6693                                        |
| Standard Deviation | 0.9559                                         | 5.2088                                         |
| Max                       | 2.2582                                         | 13.3083                                        |
| Min                       | -2.4593                                        | -15.6633                                       |

[9] Were this being released as a data product, then the issues above would be less important, because of the validation against the GLODAP dataset for example. However, this is a tool intended for users to calculate things with untested sets of input conditions. If some part of the differences between implementations are due to bugs in the code, they cannot be written off just because they're fairly small in these tests, because they could easily have a much bigger effect with a different set of inputs. In order to have confidence in the results, any unexpected behaviour or differences between implementations above the level of computer precision must be really thoroughly understood.

This is a valid point, which we believe we have addressed through addition of Appendix D.

**Code quality**

[10] I was able to get the example code to run but it still required some troubleshooting and corrections to the code beyond the instructions given in the README. These were mostly related to defining and concatenating file paths (which can more robustly and conveniently be done with os.path.join rather than by manually manipulating strings). I have made a pull request (PR) to the GitHub repo which contains these and some other (see [11]) fixes (https://github.com/LarissaMDias/PyESPER/pull/1).

Thank you for the useful comments. We have accepted and merged all aspects of this pull request.

[11] Parts of the code are very difficult to follow. This makes me worry more about points [5] and [9] above. The most critical issues are:

We agree that (as marine chemists) we have no formal training in coding and the code may be sloppy. We thank you for your careful edits! We have divided code into modules, edited all modules for clarity, and used ruff autolinter/filter (as recommended below) to check for additional errors or fixes and to make the code easier to follow.

• The functions needed are in a Jupyter notebook, so they can't be imported and used in other workflows.

We now have .py modules available. We have also completely eliminated the JupyterNotebooks from the repository.

• There are two notebooks both with copies of these functions – there should only be one "source of truth".

We now have only one copy of each function.

• Variables are defined, renamed and copied without clear reasons why, making it easy to lose track of which version of a variable should be used for the next step of the calculation.

We have closely edited code and variables within, for a more streamlined code in the final version. Additional explanations have been added for explanation of variable changes within modules.

• The deprecated seawater package is used instead of its well-maintained successor gsw.

This is done for consistency with the current MATLAB version but will be changed to the gsw package for future ESPER updates in both MATLAB and Python. When we used the gsw package within this version of PyESPER, it did not align with results from the current MATLAB version, which uses the deprecated seawater package. We have added an additional message within the PyESPER code (within the "errors.py" module) when sw is used clarifying this to users (please see below).

Error message within errors.py module of PyESPER: "Please note that, for consistency with MATLAB ESPERv1, the now-deprecated sw package is used. This will be replaced with gsw in future updates."

• It's virtually never necessary to explicitly use global variables in Python and best practice to avoid doing so.

Thank you, we have removed all unnecessary global variables.

• Numerical data appear to be processed into strings at some points?

The iterations.py module required string formatting where indexing within arrays according to string labeling (lines 160-168 of this module, see below):

Some more minor points that would improve things:

• Variables are converted between dicts and pandas DataFrames, and lists and numpy arrays, often without any clear reason. Both for code clarity and computational speed, numerical data should be kept as numpy arrays throughout, and dicts promoted to DataFrames only when essential.

Thank you for these tips. We have eliminated pandas DataFrames (replacing with dictionaries) from the package and use numpy whenever numerical data is used.

• Some packages are imported and not used (e.g., decimal).

We have eliminated these entirely.

• Some variables are defined and never used.

We have eliminated variables that were defined but not used.

• Sometimes multiple packages are used where one would be more efficient (e.g., using math and statistics for some calculations that should all be done with numpy).

Thank you for the advice. We have discontinued the math package and have eliminated use of the statistics package in all but one module ("process\_netresults.py"), where it was found to be more efficient) for our calculations in the final version.

• The code could be run through a linter / auto-styler (e.g. RuK, Black) to make it more readable and help locate some of the issues noted above.

This is a good idea, and we have used ruff linter / auto-styler for this purpose.

The PR I made to the GitHub repo (see [10]) also contains fixes for some, but not all, of the points above, and I'd be happy to discuss with the authors further on how to tackle any of these issues if that might be useful.

We thank you and have made efforts to address all of these issues.

[12] Following from [10], the authors note that the Python code runs significantly slower than the MATLAB. I suspect the frequent reliance on looping calculations through lists, which is known to be very slow in Python, rather than vectorising calculations across numpy arrays, may be largely responsible for this. Operations on pandas DataFrames can also be a lot slower than the equivalent with a dict or numpy array.

We have rewritten this section and table; many of our timing issues stemmed from using JupyterNotebooks. However, we have implemented your above comments for even greater speed. We have also updated Sect. 3.2 and Table 3 to account for changes in calculation speed for the final version.

[13] For this to be really considered "available" in Python it needs at the very least to be packaged properly and installable from the GitHub repo with pip. Functions in Jupyter notebooks are not useful for integrating into other workflows. Given my comments in [1], that this manuscript is really about the code, I think that should be a bare minimum for publication.

We thank you, this package is now installable with pip, and instructions for installing this are available in the README.

[14] Uploading to PyPI and conda-forge would be very useful additional steps, although not critical for publishing this manuscript.

We agree and are preparing to upload to both PyPI and conda-forge. This has taken a bit longer than expected due to changes in standard protocol for this process, but we hope to have this completed very soon.

**Minor comments**

[15] Figure 2: the y-axis scales have very unusual intervals, which does make it harder to interpret the figures.

We have changed the y-axis scales of Figure 2 to be much more readable, and wholenumber intervals when possible.

[16] Line 261-262: presumably "these locations" refers to the "exceptions" from the previous sentence rather than the "most ocean regions", but this is not clear.

**We have altered the language to "these exceptionally different locations"**

[17] The version number 1.01.01 is quite unusual. Of course it's the authors' prerogative to use whatever system they like, but I would suggest considering switching to the very widely used semantic versioning (https://semver.org) to make it easier to interpret.

If we understood correctly, all version numbers for this initial release should (and have been) altered to 1.0.0.

[18] For the examples, you could consider using https://github.com/mvdh7/glodap to import the GLODAP dataset (this automatically downloads the files if the user doesn't have them). I included an example script in my PR (see [10]) which shows how this could be implemented.

We thank you for the information and have included this method in our examples, rather than prior downloaded datasets.

**Responses to Reviewer #1: PyESPER**

Overall great. The first iteration of this in Matlab was already sound in my opinion so this translation requires less scrutiny. I have not run the code myself, and although it would be intensive, I believe the accessibility would improve significantly if there is a possibility for a computer scientist to create a simple UI for either packages.

We thank you for your helpful and supportive review. We hope to implement a UI in the Puget Sound in the near future that is ESPER-inspired. This could serve as a template for a more global version. We will also investigate UI solutions that can be quickly implemented as a part of this product (if time allows during this review process) or next ESPER updates, such as Voila, Mercury, Panel, or JupyterDash.

40 - Should add note of the potential high error when using a model to estimate a variable then used to calculate carbonate chemistry parameter without nutrient information too

We understood this comment to imply that ESPERs offer an alternative to these high-error model estimates and added a sentence about this following the sentence in L40 that introduces ESPERs:

L43-44. This method offers an alternative to using models to estimate variables for carbonate chemistry calculations when nutrient information is unavailable, which potentially has high error values.

50 - I would argue that it may not be considered entirely findable for many scientists who are not coding competent and even those who are, are likely unaware of the Zenodo and GitHub repositories though I recognize that is not entirely your responsibility

Yes, this is a difficult barrier. We hope to develop a simple UI that is similar to ESPERs for the Puget Sound. If successful, this could be expanded in the future for the entirety of ESPERs. We have added a sentence addressing this possibility at the end of this section for now (see below) and are testing options for easy to implement UI's to add onto this version.

L55-56. Future updates may include even more accessible features such as a user interface.

68 - If all models perform comparably then why is there a need for all three why not just use the mixed as an ensemble prediction

We have added the following information to the bottom of section 2.1 regarding this valid question:

L. 72-87. There are a couple of reasons to maintain the separate ESPER LIR, NN, or Mixed options, from an end-user perspective, and these reasons are also true for PyESPERs.

- 1. ESPER\_LIRs predate the ESPER\_NNs and have been used as a standalone data product for various research purposes (see Carter et al., 2016, doi: 10.1002/lom3.10087; Carter et al., 2018, doi: 10.1002/lom3.10232). Long-term users of these LIRs have previously expressed desire for consistency between versions (e.g., when depth was taken out as predictor for pHT), and some of them already use CANYON-B (Bittig et al., 2019) as a neural net option for comparison. Therefore, these users who desire consistency would most likely prefer to use ESPER\_LIR.
- 2. ESPER\_LIRs are more transparent than ESPER\_NN, as it is simple to parse apart coefficients at the gridded locations and easier to see how the equations are a result of these. ESPER\_LIRs also rely on a grid, which may appeal to some users.
- 3. ESPER\_NNs work a bit better on average than ESPER\_LIRs, and work more like a mapping product in that 3D coordinates are predictors, which may alternately appeal to some users.
- 4. Although the ESPER\_Mixed estimates perform better on average than LIRs or NNs do independently, there are cases where they have greater bias and RMSE than LIRs and/or NNs (e.g., when using equations 1-3 for phosphate or nitrate at all depths; Carter et al., 2021). Users may want to assess each scenario independently and choose which method is most appropriate according to their needs.
- 5. The NNs are more closely reproduced between the MATLAB and Python ESPER implementations.

100 – if there's inadequate data number and the area size is doubled, does the output indicate this? Has it been checked if this correlates with an increase in error? Why is it jumping straight to double instead of small increase intervals?

We have added the following text to help explain the rationale of the windows:

L. 122-126. In LIRv2, windows were iteratively scaled by a factor of the iteration number until at least 100 measurements are selected to train each regression. For ESPER\_LIRs (LIRv3), it is argued that increasing window size has the following benefits: (1) includes more data for regression fits, (2) introduced more modes of oceanographic variability into fitting data, and (3) reduced multicollinearity. However, the risk of increasing window size is that they will be less appropriate locally. The weighting term helps account for this (Carter et al., 2021, doi: 10.1002/lom3.10461).

Here is the weighting term used:

$$W = \max\left(5, \left(\frac{10(\Delta z)}{100 + z}\right)^2 + (\cos(\text{lat})(\Delta \text{lon}))^2 + 4(\Delta \text{lat})^2\right)^{-2}.$$

There do remain instances where the windows need to be doubled, but these amount to 5 data points out of ~50,000 (for DIC in one previous version of ESPER; no triplings of windows occurred). A previous version of ESPERs did include the data needed to determine how many doublings were required with the release for each grid cell, but we did not provide means to interpolate that information to an arbitrary location and we found that these portions of the files were rarely used. In next version updates (where we have more freedom to change the overall methods rather than replicating past ESPER methods), we hope to investigate whether doubling of window sizes has an effect on error and, if so, to modify our methods to iteratively increase window sizes instead.

160/172 - Should add a caveat that in addition to not predicting past 2030 they should not be used in areas with abnormal atmospheric CO2 absorption or profiles ie. upwelling, coastal areas, high freshwater outflow mentioned in 261 and may seem obvious to some but not others

Good point. We added the following statement:

L. 199-200. Likewise, these methods are not adequate for making reliable projections beyond the year 2030, or perhaps sooner in coastal or other areas where the underlying global open-ocean anthropogenic carbon estimations have greater uncertainties.

**Responses to Reviewer #2: PyESPER**

The manuscript describes a new python-based version of the existing ESPER algorithms. No new development or training is performed, but detailed comparison of outcomes with both the original Matlab and the new Python versions is described.

The manuscript is well written and clearly details what is new and how the new version performs compared to the original. It is a nice added value that the algorithms are now available in several programming languages.

I only have some minor comments. Once those issues are fixed I'm happy to see this published.

We thank you for your productive feedback.

Minor issues:

Throughout the manuscript information is needlessly repeated several times. In particular which observational data are included is presented again and again. It should be sufficient to define once what is o and w data, including how much data there is, and then just refer to those definitions. The many repetitions of this information makes the text a bit cumbersome to read.

We have edited the manuscript for repeated information and deleted duplicates of data definitions, including the following:

We removed the (repeated from prior text) number of measurements within GLODAPv2.2022 within the caption of Fig. 1 (L251), definition of open ocean data from captions of Table 1 (L284), Fig. 2 (L297), Fig. 3 (L343), Table 2 (L395), Fig. 7 (L409), and definition of whole ocean data from the captions of Table B1, Fig. B1 (L497), Fig. B2 (L509), Fig. B3 (L515), Table B2 (L524), and Fig. B4 (L537).

L261. Redefinition of open ocean data was removed.

Line 141: ensemble is the more commonly used word so I suggest using only that

The wording has been changed to ensemble, with no mention of committees (now L169).

Line 156-158: The sentence is quite awkwardly phrased. Try revising for clarity.

True. The wording of the entire paragraph has been revised for clarity as follows:

L184-188. The impacts of anthropogenic carbon (Cant) are approximated in ESPER and PyESPERv1.0 using a 1° x 1° gridded transit time distribution (Waugh et al., 2006)-based Cant product referenced to the year 2002 (Lauvset et al., 2016). ESPERs assume that oceanic Cant increases proportionally to atmospheric anthropogenic CO2 (transient steady state assumptions; Gammon et al., 1982; Gruber et al., 2019; Tanhua et al., 2007). This implies that the "shape" of the Cant vertical profile (gradient) remains constant with continuous exponential increases of atmospheric CO2 and ocean Cant according to Eq. (3; Carter et al., 2021).

Line 160: I do not understand the meaning of the sentence. Please revise for clarity.

We have reworked the entire paragraph for clarity (see above Line notes).

Line 230-235: All these numbers are also given in the table so it is unnecessary to repeat here. The information is also more easily digestible from a table. Same goes for lines 275-279.

We have replaced these two segments of text with simple reference to Tables as follows:

L280-281. Mean (±standard deviation; RMSEn) PyESPER – ESPER\_LIR differences for each property are shown in Table 1.

L338-339. Mean (±standard deviation; RMSEn) offset for each property is shown in Table 2.

Table 2 caption: Try splitting the information into smaller sentences or removing some redundant information.

**We have reworked the caption as follows:**

L414-417. Table 2: Mean (standard deviation), maximum, minimum, and normalized RMSE (RMSEn) are shown for three scenarios: (1) between Python – MATLAB NNs, (2) MATLAB ESPER\_NN – measured values, and (3) PyESPER\_NN – measured values. Separate rows exist for TA, DIC, pHT, phosphate, nitrate, silicate, and oxygen estimates. All units except pHT are  $\mu$ mol kg-1, and data are for open oceans ( $_0$ ) and all equations combined.

Captions for Figures 2, 7, B1 and B4: There is no information about what the histograms/bars on the top and right represent. This should be added.

We have added the following sentence to each of these figure captions:

Top and bottom side histograms represent the distribution of the x and y axes, respectively.

Lines 287-288: This statement appears to contradict the information on lines 318-319. Please clarify.

Yes, this is confusingly worded. We have clarified the point we were trying to make, which was that, despite some minor offsets in  $C_{ant}$  estimates for pH $_{\rm T}$  and DIC due to interpolation differences, estimates from NNs for these two variables remain functionally identical. Please refer to the following changes in-text.

L346-347. These minor offsets are attributed to the programming language differences in the interpolation of the Cant adjustment, which is only applied to these two properties.

L395-396. Currently, when Cant estimates are required, the results from PyESPER\_NNs remain functionally identical to those from ESPER\_NNs, despite minor offsets from the interpolation methods.

Figure 4: Most differences are found in the northwest Pacific Ocean. It would be interesting if you could add a brief discussion about why this is and the implications of it.

The reason for these discrepancies within the western Pacific Ocean is that this is a place where GLODAPv2.2022, which was used for estimate comparisons, contained data from the very deep ocean. Very deep locations are at or near the edge of the original MATLAB grid for training data (5500 m), where interpolation methods had greater differences. You

can see the matching "problem areas" on the following (coarse) map of locations where GLODAPv2.2022 samples were collected at >6000 m depth:

We have added the following text to the text on LIR results to better explain this:

L324-329. PyESPER\_LIRs were within  $2\sigma$  (~95% of measurements should fall within this uncertainty level) for most ocean regions, with a few exceptions which occurred predominantly in coastal areas or deep waters near the edges of the original MATLAB grid (Figs. 3 and 4). Spatial patterns in distribution of outliers shown in Fig. 4 appear to reflect locations where more edge-of-grid biogeochemical measurements were collected (e.g., near coasts and in deep waters). Hence, these locations aligned well with places where coefficients were extrapolated in the MATLAB implementation (see Sect. 2.1.1, "Locally interpolated regressions"; Figs. 3, 4, and 5; for  $_{\rm w}$  Fig. B2 and B3).

Line 320: I suggest you rename this section. It is not intuitive that it deals with the differences in speed of calculation

We have renamed the section to "Speed of calculation," as suggested.

Figure 6: It would be useful to have a panel showing the differences between panels a and b

This has been added to Figure 6, as recommended. This figure was moved to appendices (now Fig. C1), as it is likely more suited there. Please see the following modified figure:

Line 358: I suggest to rename the section future work or future improvements.

**We have renamed the section "Future improvements"**

Please also list the data product doi(s) in the data availability section along with the references.

We have included the doi's for the appropriate data repositories in this section.

Line 383-384: "essentially identical" is not true for DIC and pH. That should be mentioned here too

The estimates are still very closely aligned for DIC and  $pH_T$  when estimated using NN methods ( $C_{ant}$  contributions account for a slight difference between the MATLAB and Python estimates). We have noted that here and removed the "essentially identical" language as follows.

L489-490. Estimates from PyESPER\_NNs precisely align with those from ESPER\_NNs for all equations and desired outcome variable combinations (Fig. 7) and estimates from these two routines align very closely for all estimates, and to within machine precision for all but pH $_{\rm T}$  and DIC, which exhibit slight differences due to impacts of interpolating for  $C_{\rm ant}$ .

Table A1: Caption refers to Table S2, but that is really A2.

Thank you for pointing this out. We have fixed this error.

**Responses to Reviewer #3: PyESPER**

This manuscript introduces PyESPERv1.01.01, a Python-based implementation of empirical seawater property estimation routines (ESPERs), previously developed and made available only in MATLAB by author Carter. These routines estimate core seawater biogeochemical properties —such as total alkalinity, dissolved inorganic carbon, total pH, nitrate, phosphate, silicate, and oxygen—using inputs like geographic coordinates, depth, salinity, and up to four additional predictors (e.g., temperature and biogeochemical information). Two statistical algorithms, a locally interpolated regression (LIR) and a neural network (NN) estimation are averaged to produce a best estimate.

By transitioning ESPERs to Python, the authors enhance accessibility for the scientific community, as Python is an open-source language widely used in oceanographic research. The study also documents modifications made to reduce discrepancies between the Python and MATLAB implementations and evaluates the disagreements between the methods. The implementation also updates underlying datasets using Global Ocean Data Analysis Project (GLODAPv2.2022) dataset and addresses a couple minor issues with the original code.

The work submitted here will be a valuable resource to the community and required a large amount of detailed assessment and validation. I recommend publication after consideration and edits based on the range of suggestions from reviewers.

**We thank you for the constructive feedback.**

**General Feedback:**

This work will have substantial impact on the field of ocean biogeochemistry and carbon cycling, as well as serve as an important resource for characterizing baseline inorganic carbon chemistry in the context of marine carbon dioxide removal (mCDR) activities. While the

concepts and ideas are not new, and build on the original ESPER, transitioning this tool to Python will broaden accessibility and encourage further scientific inquiry and discovery.

**We concur and appreciate the feedback**

The calculations/algorithms used are described in precise and comprehensive detail. Care is taken to evaluate uncertainty, as well as assess internal consistency within the inorganic carbon system.

**Thank you for the feedback.**

I commend the authors for making the code available on GitHub through a Jupyter Notebook example. However, two improvements would make this much more accessible to the community: (1) I am very surprised the performance was so much worse with python relative to Matlab. Profiling the code to see where the slowdown is likely could lead to massive performance improvements with some refactoring. (2) providing the code in a pip or conda installable package would make it much more reproducible and less error prone.

We appreciate the suggestions for improvements. (1) We have indeed profiled the code and found the slowdown to be during interpolations. This was greatly improved by packaging it, which we are near completion of. (2) We are nearing completion of the pip installable package also, which should be ready by the time of formal publication. Please check the GitHub page for the package.

The overall presentation is clear, although somewhat dense. I appreciate the detailed documentation of methodology though.

**Thank you for the comments.**

**Minor Feedback:**

Do you have insight into why DIC and pH seems to have considerably larger python-Matlab differences?

Yes, this is because the current methods for estimation of contributions of anthropogenic carbon ( $C_{ant}$ ) to DIC and pH involves interpolations, which did not match well between Python and MATLAB versions. Other estimated properties (e.g., TA, nitrate, phosphate, silicate, and oxygen) do not require estimates of  $C_{ant}$ . Please see the following modified explanation to help clarify this.

L. 347-349. The largest relative disagreements were found for DIC and pH $_{\rm T}$ , though these disagreements remained small relative to measurement uncertainties. These minor offsets are attributed to the programming language differences in the interpolation of the  $C_{ant}$  adjustment, which is only applied to these two properties.

L145: For clarification – NN functions were translated from scratch? Was this compared to using something 'out of the box' like pytorch? It would be interesting to compare both reproducibility and performance.

We did translate the neural networks from scratch because we wanted an exact replica (to the best of our ability). The translation (PyESPER\_NN) indeed did replicate ESPER\_NN results to within machine precision. We feel that it is unlikely that independently trained neural networks would provide as similar results as our present method, but do not rule out the possibility of providing a "python-trained" option in future ESPER updates.

Figure B2: There seems to be structure in the large mismatches – For example in the North Pacific along margins, and perhaps on an A10 GO-SHIP line. Could you add discussion on this? Does this point towards potentially a data problem with one cruise?

This is true that there are areas where the mismatch are greater (although not for one particular cruise). These areas align with places where the "edges" of our interpolated grid occur. This is caused by differences in interpolation and extrapolation between the two coding languages, where interpolating between previously extrapolated areas (in MATLAB) is not a very good reproduction of the MATLAB mathematical method. We have modified the text as follows, to aid with this explanation.

L. 324-330. PyESPER\_LIRs were within  $2\sigma$  (~95% of measurements should fall within this uncertainty level) for most ocean regions, with a few exceptions which occurred predominantly in coastal areas or deep waters near the edges of the original MATLAB grid (Figs. 3 and 4). Spatial patterns in distribution of outliers shown in Fig. 4 appear to reflect locations where more edge-of-grid biogeochemical measurements were collected (e.g., near coasts and in deep waters). Hence, these locations aligned well with places where coefficients were extrapolated in the MATLAB implementation (see Sect. 2.1.1, "Locally interpolated regressions"; Figs. 3, 4, and 5; for  $_{\rm W}$  Fig. B2 and B3). Within regions where MATLAB was interpolating, far outliers were uncommon (Figs. 3, 4, 5, B2, and B3).

Figure B3: The colorbar should 'depth' but there are no labels or units?

The labels and units appear to the right of the figure (please see below).